# Strigolactones are chemoattractants for host tropism in Orobanchaceae parasitic plants

Satoshi Ogawa [1], Songkui Cui [2], Alexandra R. F. White [3], David C. Nelson [3], Satoko Yoshida[2,4] & Ken Shirasu [1,5] ✉

Parasitic plants are worldwide threats that damage major agricultural crops. To initiate infection, parasitic plants have developed the ability to locate hosts and grow towards them. This ability, called host tropism, is critical for parasite survival, but its underlying mechanism remains mostly unresolved. To characterise host tropism, we used the model facultative root parasite *Phtheirospermum japonicum*, a member of the Orobanchaceae. Here, we show that strigolactones (SLs) function as host-derived chemoattractants. Chemotropism to SLs is also found in *Striga hermonthica*, a parasitic member of the Orobanchaceae, but not in non-parasites. Intriguingly, chemotropism to SLs in *P. japonicum* is attenuated in ammonium ion-rich conditions, where SLs are perceived, but the resulting asymmetrical accumulation of the auxin transporter PIN2 is diminished. *P. japonicum* encodes putative receptors that sense exogenous SLs, whereas expression of a dominant-negative form reduces its chemotropic ability. We propose a function for SLs as navigators for parasite roots.

Plant parasitism has independently evolved at least 12 times, accounting for about 1% of angiosperms or about 4500 species[1,2]. A key trait common to parasitic plants is the ability to connect to host plants and deprive them of nutrients and water, which is often harmful to the hosts[3,4]. Among parasitic plants, some Orobanchaceae species such as *Striga* spp. and *Orobanche* spp. are notoriously devastating invaders of major agricultural crops, leading to a multibillion-dollar economic loss annually[5]. Most obligate Orobanchaceae parasites have evolved three steps to complete invasion[2]: germination near the host root in response to host-derived stimulants such as strigolactones (SLs)[2], active extension of the parasite's root to host roots[6–8], and connection of the vasculature via formation of an invasive organ called a haustorium[2,9–12]. Although many studies have focused on germination and haustorium development, the molecular basis for host tropism is largely unknown.

*Phtheirospermum japonicum* is a facultative hemiparasite in the Orobanchaceae that completes its lifecycle with or without a host[13,14].

With the advent of genetic tools, including ethyl methyl sulfonate-driven mutagenesis and *Agrobacterium rhizogenes*-mediated root transformation[10,15,16], *P. japonicum* has been used as a model Orobanchaceae species to investigate parasitism. These studies have revealed the molecular mechanisms regulating haustorium development, vasculature connection and exchange of molecules[12–14,16–20]. As a facultative parasite, *P. japonicum* does not require host-derived stimulants, such as SLs, for germination. Hence, we considered *P. japonicum* to be an ideal plant for studying tropic mechanisms separate from germination.

In angiosperms, α/β hydrolases DWARF14 (D14) act as SL receptors[21,22]. After SL recognition, D14 interacts with an F-box protein MORE AXILLARY GROWTH2 (MAX2), a component of a SKP1-CULLIN-F-BOX (SCF) E3 ubiquitin ligase complex (SCF$^{MAX2}$), to mediate SL signalling[23]. Interestingly, SCF$^{MAX2}$ also mediates signalling driven by karrikins (KARs), a family of compounds found in smoke that regulate plant growth[24,25]. In *Arabidopsis*, responses to KARs require *KARRIKIN*

[1]RIKEN Center for Sustainable Resource Science, Yokohama 230-0045, Japan. [2]Division of Biological Science, Graduate School of Science and Technology, Nara Institute of Science and Technology, Ikoma, Nara 630-0192, Japan. [3]Department of Botany and Plant Sciences, University of California, Riverside, CA 92521, USA. [4]PRESTO, Japan Science and Technology Agency, Kawaguchi, Saitama 332-0012, Japan. [5]Graduate School of Science, The University of Tokyo, Tokyo 113-0033, Japan. ✉e-mail: ken.shirasu@riken.jp

*INSENSITIVE 2* (*KAI2*)/*HYPOSENSITIVE TO LIGHT* (*HTL*) (hereafter *KAI2*), a paralogue of D14[26]. Whereas *KAI2* is a single gene in *Arabidopsis*, Lamiid species such as Solanales and Lamiales have duplicated *KAI2* genes. KAI2 proteins in Lamiids are categorised into three types: a "conserved" type found in most Lamiids as well as other angiosperms (KAI2c), an "intermediate" type only found in Lamiales, including

parasites and non-parasites with a few exceptions such as *Orobanche* spp. (KAI2i), and a "divergent" type conserved uniquely in Orobanchaceae parasites (KAI2d)[27,28]. Phylogenetic analyses revealed that Orobanchaceae parasites have rapidly duplicated the *KAI2d* genes, resulting in multiple genomic copies[27–29]. Some facultative Orobanchaceae parasites that do not require host-derived SLs to germinate

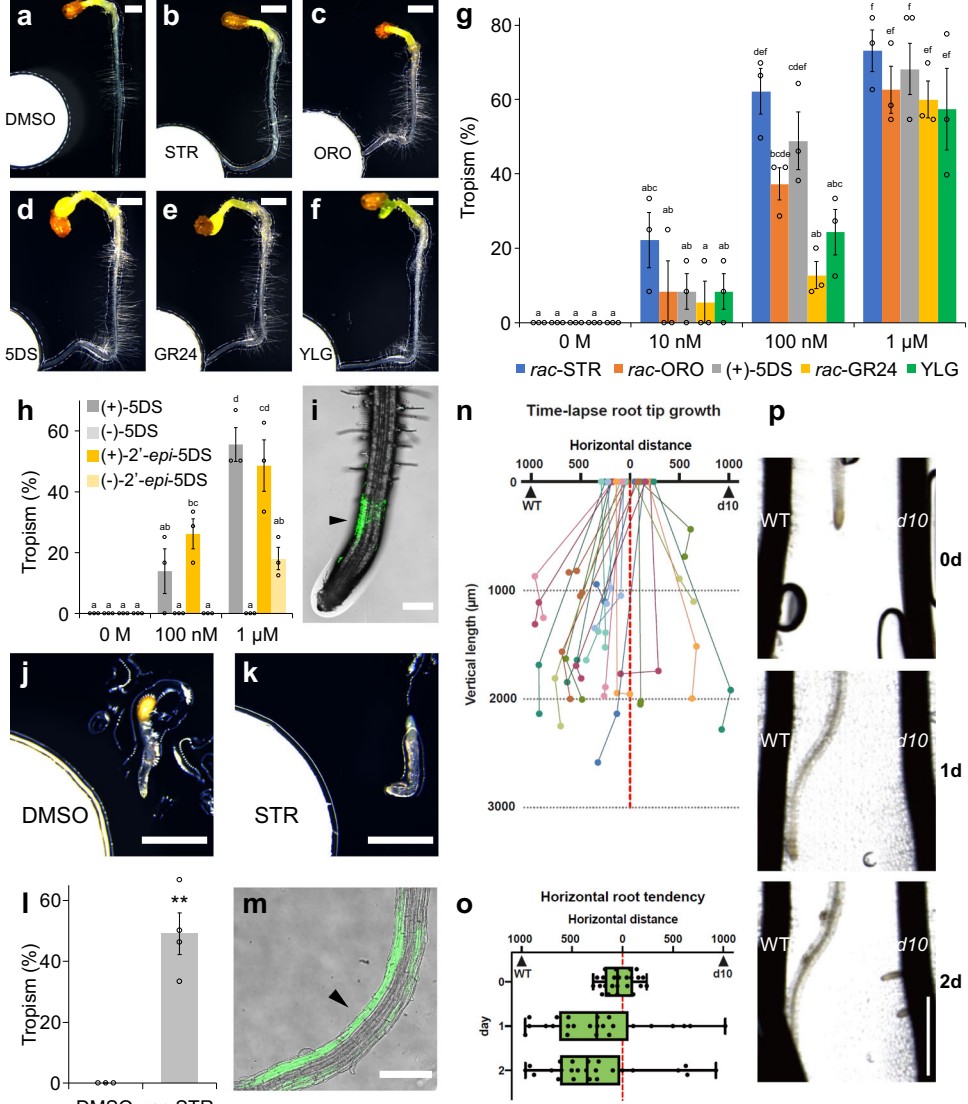

**Fig. 1 | Chemotropic phenotype of Orobanchaceae parasitic plants to SLs and analogues. a–f** Representative images of *P. japonicum* seedlings treated with 1 μM chemical solutions diluted in 0.1% DMSO. Photos were taken 1 day after treatment. **a** 0.1% (v/v) DMSO; **b** *rac*-strigol (STR); **c** *rac*-orobanchol (ORO); **d** (+)-5DS; **e** *rac*-GR24; **f** YLG. **g**, **h** Percentage of *P. japonicum* plants that showed chemotropism to each chemical. Three independent batches (3–12 and 4–8 seedlings in each batch for **g** and **h**, respectively) for each compound. **i** A representative image of *P. japonicum* exhibiting YLG-derived fluorescence after treatment with a 100 μM YLG solution. Filter paper disks were placed 5-mm to the left of the roots. Confocal photos were taken 6 h after treatment. **j**, **k** Representative images of *S. hermonthica* seedlings treated with 1 μM *rac*-strigol or 0.1% (v/v) DMSO solutions. Photos were taken 1 day after treatment. Eighteen out of 22 plants showed a similar fluorescent pattern. **j** 0.1% DMSO; **k** *rac*-strigol. **l** Percentage of *S. hermonthica* plants that showed chemotropism to 1 μM *rac*-strigol. Three or four independent batches (2–13 plants) were treated with each compound. **m** A representative image of *S. hermonthica* exhibiting YLG-derived fluorescence when treated with a 100 μM YLG solution. Filter paper disks were placed 3-mm to the left of the roots. Confocal photos were taken 24 h after treatment. Thirteen out of 16 plants showed a similar

fluorescent pattern. **n–p** Root chemotropism experiments in *P. japonicum* towards intact rice. Each *P. japonicum* root was placed between a WT root and a *d10* root that were aligned vertically at ~2-mm apart. Quantitative measurements (**n**, **o**) and images (**p**) were collected at 0, 1 and 2 day(s) post co-incubation. **n** Time-lapse growth of *P. japonicum* roots. Plots indicate the positions of each root. **o** Horizontal distances between *P. japonicum* root tips and rice roots. Lower quartile, upper quartile, and median are presented as left corner, right corner, and central line of the box, respectively. Left and right edges of whiskers indicate minima and maxima, respectively. **p** Representative images of a *P. japonicum* root growing towards WT and not towards *d10*. Experiments were repeated three times with a total of 24 *P. japonicum* seedlings. **g**, **h**, **l** Mean ± standard error of the mean (SEM). Roots in which growth had stopped were excluded from the calculation. **i**, **m** Arrowheads indicate asymmetrical fluorescence. **g**, **h** Different letters indicate statistical significance at $p < 0.05$ (two-way ANOVA, Tukey's multiple comparison test). **l** **$p < 0.01$ (Welch's *t* test, two-sided). Scale bars indicate 1 mm for **a–f**, **j**, **k**, **p**, and 200 μm for **i**, **m**, respectively. **g**, **h**, **l** experiments were performed at least three times with similar results. Source data provided.

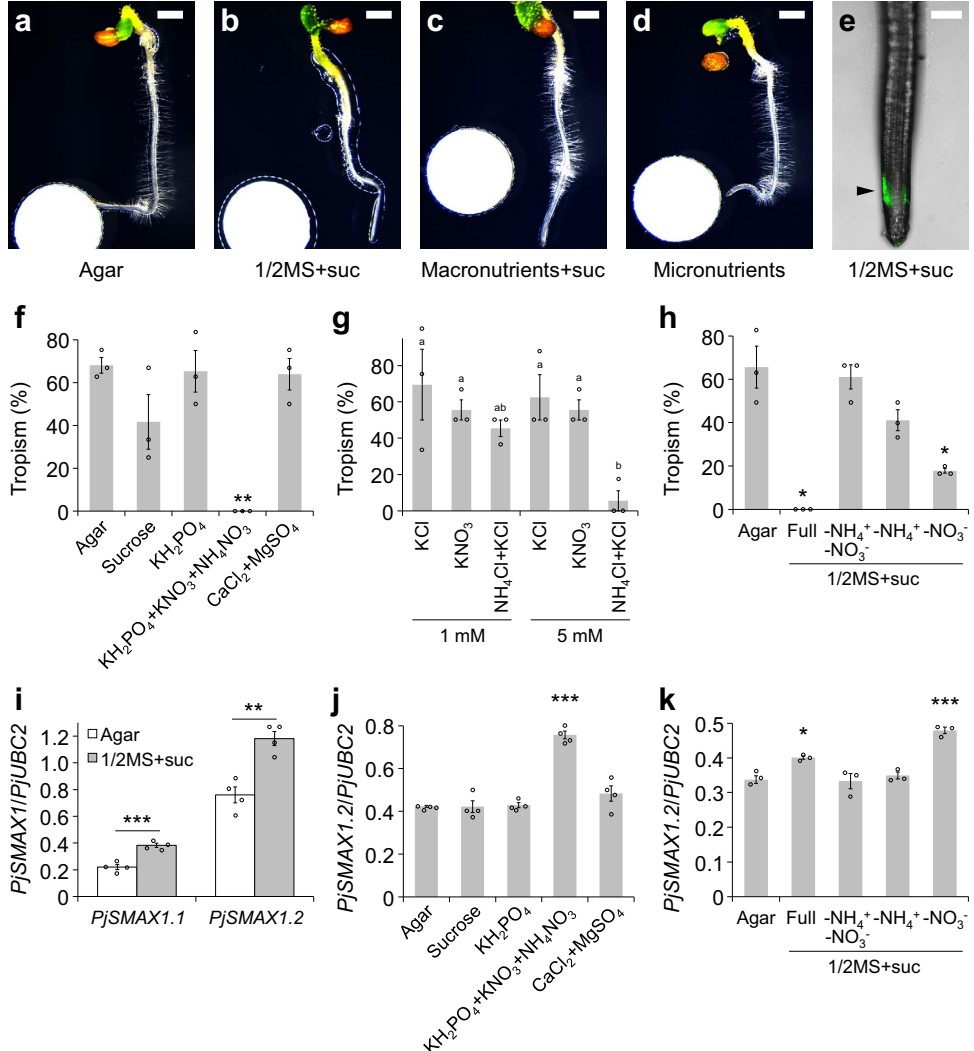

**Fig. 2 | Chemotropic phenotype towards SLs on nutrient-containing media.**
**a–d** Chemotropic responses to *rac*-strigol in varying nutrient conditions in the presence of a 1 μM *rac*-strigol solution. Photos were taken 1 day after treatment. **a** agar without any supplementary nutrients; **b** 1/2MS (Nihon Pharmaceutical) with 1% (w/v) sucrose (suc); **c** macronutrients of 1/2MS + suc; **d** micronutrients of 1/2MS without suc. **e** A representative image of *P. japonicum* plants showing YLG-derived fluorescence when grown on 1/2MS + suc upon treatment with 100 μM YLG solution. Filter paper disks were placed 5-mm to the left of the roots. Confocal photos were taken 6 h after treatment. The arrowhead indicates asymmetrical fluorescence. **f–h** Percentage of *P. japonicum* exhibiting chemotropism to 1 μM *rac*-strigol on each medium. Three independent batches (3–12 plants) were tested for each compound. Plants in which root growth had stopped were excluded from the calculation. **f** Assays on agar containing varying macronutrients or sucrose. **g** Assays on agar and a reduced nitrogen source. KCl was added to adjust the

potassium concentration. **h** Assays on 1/2MS + suc agar with a limited nitrogen source. **i–k** Relative expression level of *PjSMAX1* determined by RT-qPCR. Representative data are shown using *PjUBC2* as the reference gene. **i** Expression levels of *PjSMAX1.1* and *PjSMAX1.2* on agar without nutrients or 1/2MS + suc (four technical replicates). **j** Expression level of *PjSMAX1.2* on agar with each macronutrient or sucrose (four technical replicates). **k** Expression level of *PjSMAX1.2* on 1/2MS + suc agar with a limited nitrogen source (three technical replicates). **f–k** Mean ± SEM. **f**, **h**, **j**, **k** *$p < 0.05$, **$p < 0.01$, ***$p < 0.001$ (Welch's *t* test, two-sided) in comparison with the no nutrient treatment. **g** Different letters indicate a statistical significance at $p < 0.05$ (two-way ANOVA, Tukey's multiple comparison test). **i** **$p < 0.01$, ***$p < 0.001$ (Welch's *t* test). Scale bars indicate 1 mm for **a–d** and 200 μm for **e**, respectively. **f–k** Experiments were performed three times with similar results. Source data provided.

also encode multiple KAI2d proteins, suggesting that KAI2d may have a yet uncovered function distinct from germination stimulants.

Here, we show that *P. japonicum* roots exhibited chemotropism to various SLs. Consistent with this result, *P. japonicum* roots tended to grow towards a wild-type (WT) host rather than an SL-deficient mutant host. Chemotropism to SLs was also observed in *Striga hermonthica*, but not in non-parasitic plants, suggesting that this strategy is potentially Orobanchaceae parasite-specific. Chemotropism to SLs in *P. japonicum* is negatively regulated by ammonium in the medium, implying that parasites infect hosts when the nitrogen source is limited. We also describe a functional link between SLs and the plant hormone, auxin, in chemotropism. Thus, our findings provide a function for SLs as host-produced chemoattractants for parasites.

## Results

### Orobanchaceae parasitic plants exhibit chemotropism to SLs

We previously identified multiple homologues of SL receptor-encoding genes in the genomes of *P. japonicum* and other Orobanchaceae parasites[27], although *P. japonicum* does not require exogenous SLs for germination. Therefore, we hypothesised that *P. japonicum* might use SLs as chemoattractants rather than for germination. To evaluate this putative function, we devised an experimental system in which a chemical-soaked filter paper disk was placed next to seedlings. Dimethyl sulfoxide (DMSO) at a concentration of 0.1% (v/v) was used as the negative control. Using this assay system, we demonstrated that *P. japonicum* exhibited chemotropism to three SLs (*rac*-strigol, *rac*-orobanchol and (+)-5-deoxystrigol (5DS)) and two synthetic analogues

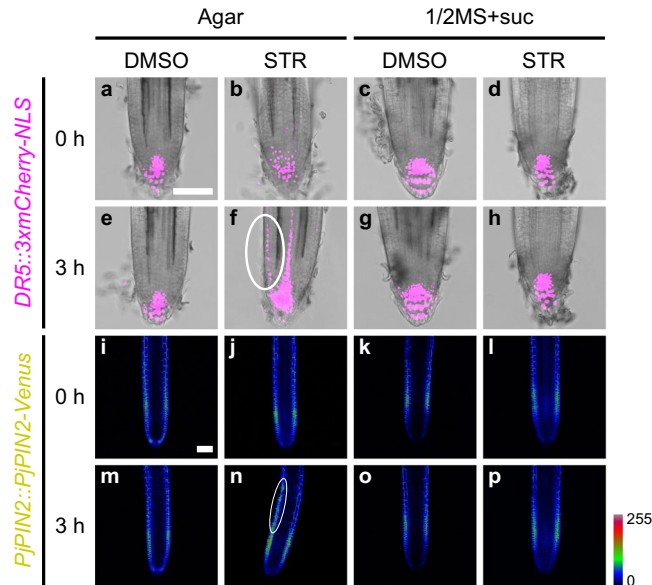

**Fig. 3 | Expression dynamics of auxin-related genes upon SL treatment.**
Expression patterns of the *DR5* (**a**–**h**) and *PjPIN2* (**i**–**p**) promoters driving expression of a fluorescent marker gene upon treatment with 1 μM *rac*-strigol (STR) or 0.1% (v/v) DMSO solutions. Filter paper disks were placed 5-mm to the left of the roots. Confocal photos were taken at the indicated time points after treatment on media with (1/2MS + suc) or without (agar) nutrient. Bright-field and fluorescent images were merged in **a**–**h**. Venus fluorescence intensity was represented in a Rainbow RGB spectrum in **i**–**p** using ImageJ[71]. White ovals indicate asymmetrical expression. The scale bars in **a** and **i** also refer to **b**–**h** and **j**–**p**, respectively. Bars = 100 μm. At least six biological replicates were performed for each condition with similar results.

(*rac*-GR24 and Yoshimulactone Green (YLG)[30] (Fig. 1a–f and Supplementary Fig. 1 and Supplementary Movie 1). Among the tested chemicals, *rac*-strigol and (+)-5DS had the strongest chemotropism-inducing activity at a concentration of 100 nM (Fig. 1g). We also compared the chemotropism activities between stereoisomers of 5DS (Supplementary Fig. 1), as the chemical structure of SLs is important for bioactivities such as germination-inducing activity and hyphal branching-inducing activity[31–33]. In contrast to (+)-5DS, the unnatural isomer (−)-5DS did not induce chemotropism (Fig. 1h). The chemoattractant activity of (+)-5DS was comparable to that of (+)-2′-*epi*-5DS, an SL found in rice[34], and stronger than another unnatural isomer (−)-2′-*epi*-5DS (Fig. 1h). This result suggests that *P. japonicum* can efficiently sense host-derived natural SLs for chemotropic activity. In addition, using YLG, which is an indicator of SL receptor activity by binding to SL receptors and being hydrolysed into fluorescent products[30], we observed asymmetrical fluorescence in the root elongation zone (Fig. 1i), suggesting that asymmetrical SL recognition activity induces an asymmetrical root elongation pattern, leading to chemotropism. To investigate the chemoattractant activity of SLs on another parasitic plant in the Orobanchaceae family, we used *S. hermonthica* seedlings just after germination. We found that *S. hermonthica* exhibited chemotropism to *rac*-strigol and had an asymmetrical YLG recognition pattern like *P. japonicum*; however, growth was limited probably due to the lack of sufficient nutrients in the typically small seeds (Fig. 1j–m). By contrast, a non-parasitic Orobanchaceae *Lindenbergia philippensis* exhibited neither chemotropism to *rac*-strigol nor asymmetrical YLG recognition (Supplementary Fig. 2a, b). *Arabidopsis thaliana*, a non-parasitic plant in the Brassicaceae, also did not show chemotropism to *rac*-strigol (Supplementary Fig. 2c, d). These data imply that chemotropism to SLs might be limited to Orobanchaceae parasitic plants.

Next, we conducted infection assays using *P. japonicum* and rice to test the effect of host-derived SL on host tropism. In agreement with the chemotropism results, more *P. japonicum* roots grew towards WT

rice roots than *dwarf10* (*d10*), an SL-deficient mutant that lacks one of the SL-biosynthetic genes[35] (Fig. 1n–p and Supplementary Movie 2). Therefore, we concluded that host-derived SLs are important for host tropism in *P. japonicum*. The chemotropic response in hemiparasitic plants appear to be specific to SLs, as karrikins (KAR1 and KAR2) and costunolide, a sunflower chemoattractant for Orobanchaceae holoparasite *Orobanche cumana*[8], did not exhibit chemotropic activity for *P. japonicum* and *S. hermonthica* at a concentration of 1 μM (Supplementary Fig. 3).

## Ammonium ions are important for regulating chemotropism
Improvement of soil fertility blocks the proliferation of *Striga* spp.[36]. Thus, we considered that tropic responses to SLs, one strategy for parasitism, may also be compromised by the presence of abundant nutrients. To test this hypothesis, we evaluated chemotropism to *rac*-strigol on half-strength Murashige-Skoog (1/2MS) agar with sucrose[37], a representative nutrient-rich medium on which *P. japonicum* can grow without parasitism[14]. In contrast to assays on water agar, chemotropism to *rac*-strigol was compromised on 1/2MS agar with sucrose (Fig. 2a, b). We also noticed that chemotropism to *rac*-strigol was inhibited on agar containing MS macronutrients (KH$_2$PO$_4$, KNO$_3$, NH$_4$NO$_3$, CaCl$_2$ and MgSO$_4$) and sucrose, but not agar containing MS micronutrients, (Fig. 2c, d and Supplementary Fig. 4). In contrast, asymmetrical YLG-derived fluorescence remained in the root elongation zone (Fig. 2e), indicating that nutrients affect the signalling pathway downstream of the SL receptors but not the perception process. To identify the component(s) that impairs chemotropism to SLs, we conducted chemotropism to *rac*-strigol assays using water agar supplemented with individual MS macronutrients or sucrose (Fig. 2f). Note that we added KH$_2$PO$_4$ to nitrogen (KNO$_3$ and NH$_4$NO$_3$)-containing media in which the nutrient concentration was equal to 1/2MS because the addition of only a nitrogen source to these media was toxic to *P. japonicum*. We found that nitrogen, especially ammonium ions, significantly compromised the chemotropic response to *rac*-strigol (Fig. 2f, g). Since acidic stress is known to cause ammonium toxicity in *Arabidopsis*[38], we tested the effect of ammonium ions on chemotropism to SLs at a neutral pH condition, only to obtain a similar result (Supplementary Fig. 5). When ammonium ions were omitted from 1/2MS with sucrose, chemotropism activity was recovered, demonstrating that ammonium ions, but not sucrose, are necessary and sufficient to impair chemotropism to SLs (Fig. 2h). Next, to investigate downstream signalling, we focused on *SUPPRESSOR of MAX2 1* (*SMAX1*) genes in *P. japonicum* that encode homologues of a negative regulator of SL signalling[39,40]. As expected, expression of *PjSMAX1* was enhanced on 1/2MS and media containing ammonium ions, suggesting that SL signalling to trigger chemotropism was suppressed (Fig. 2i–k). Note that we selected PjSMAX1.2 for the analyses, because PjSMAX1.1 lacks several amino acid sequences conserved in other SMAX1/SMXL2 proteins in Lamiales and it might not be functional (Supplementary Fig. 6). Consistently, overexpression of *PjSMAX1.2* resulted in loss of chemotropism to *rac*-strigol (Supplementary Fig. 7). Overall, our data suggest that *P. japonicum* negatively regulates the SL signalling pathway in response to ammonium ions, leading to a reduction in host tropism capacity. We inferred from the previous studies on homologues of PjSMAX1.2[25,41] that SL signalling might trigger the degradation of PjSMAX1.2 protein, but ammonium ions might counteract the effects of the degradation by enhancing *PjSMAX1.2* expression. A mechanism for SL and cytokinin cross-talk via a homologue of PjSMAX1.2 in pea has been proposed[42]. Hence, the downstream signalling of SL perception, indicated by YLG-derived fluorescence, might be affected by the signalling via endogenous cytokinin.

## Auxin response contributes to the tropism to host
Auxin is a phytohormone well known to regulate tropisms, as exemplified by gravitropism[43]. To test whether an auxin response also regulates chemotropism to SLs, we used *P. japonicum* hairy roots

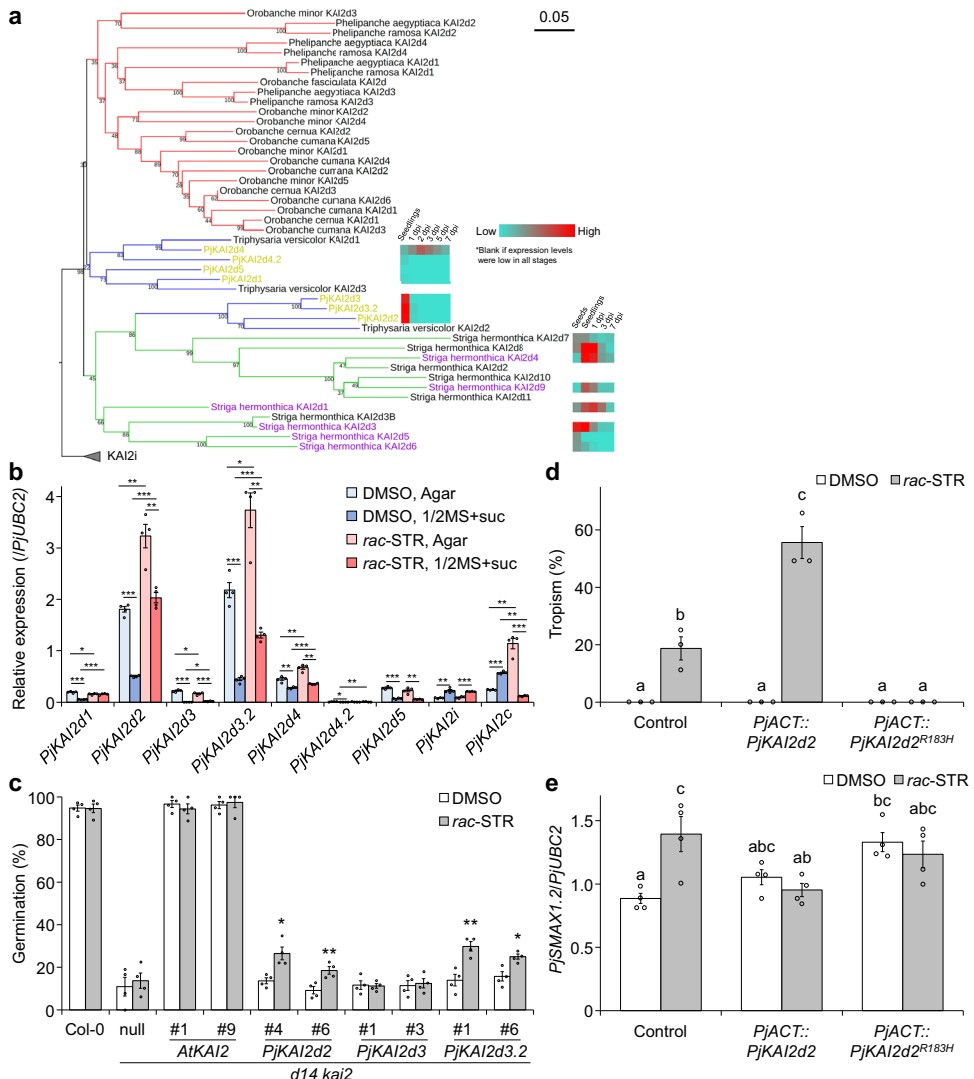

**Fig. 4 | Analyses of KAI2 homologues in *P. japonicum*. a** KAI2d phylogeny. The KAI2i clade was included as an outgroup. Clades are coloured blue, green, and red for facultative hemiparasites, obligate hemiparasite, and obligate holoparasites, respectively. Purple and yellow words indicate KAI2d forms in *S. hermonthica* that recognise SLs[30, 77] and KAI2d forms in *P. japonicum* (PjKAI2d), respectively. Coloured boxes indicate gene expression levels in roots of *P. japonicum* for *PjKAI2d* or in seeds, seedlings, or rice-infecting plants at 1-, 3-, 7-day post infection (dpi)[18, 29]. The bar indicates substitutions per site. Bootstrap values are indicated at the nodes. **b** Relative expression levels of *PjKAI2* genes. Representative data are shown (four technical replicates) using *PjUBC2* as the reference gene. **c** Complementation assays for PjKAI2d. *Arabidopsis kai2 d14* (Col-0 ecotype) was used for the null mutant to express PjKAI2d under control of the *AtKAI2* promoter. Germination rates were calculated 5 days after incubation at 22 °C in the dark. Representative data are shown (four independent batches, 23–87 seeds per batch). Two independent lines were selected and tested for each transgene. **d, e** Chemotropic responses to *rac*-strigol (STR) in *PjKAI2d2* dominant negative plants. **d** Percentage of *P. japonicum* transgenic hairy roots that were chemotropic to 1 μM *rac*-strigol or 0.1% (v/v) DMSO. Three independent batches (3–9 plants) for each compound. Plants that stopped root growth were excluded from the calculations. **e** Relative expression level of *PjSMAX1.2*. Representative data are shown (four technical replicates) using *PjUBC2* as the reference gene. **b–e** Mean ± SEM. **b** \**p* < 0.05, \*\**p* < 0.01, \*\*\**p* < 0.001 (Welch's *t* test, two-sided). **c** \**p* < 0.05, \*\**p* < 0.01, \*\*\**p* < 0.001 (Welch's *t* test, two-sided), in comparison to DMSO treatment for each genotype. **d, e** Different letters indicate a statistical significance at *p* < 0.05 (two-way ANOVA, Tukey's multiple comparison test). **b–e** Experiments were performed three times with similar results. Source data provided.

transformed with the 3X mCherry-nuclear localisation signal (NLS) module driven by the auxin-responsive promoter *DR5*[12,19]. We found that asymmetrical activation of the auxin response in root tip epidermal cells was towards *rac*-strigol on water agar. This asymmetrical activation was diminished on 1/2MS medium containing sucrose (Fig. 3a–h). Given the results shown in Fig. 2, it is reasonable to suggest that MS macronutrients compromise the asymmetric auxin response rather than sucrose. This strigol- and starvation-inducible asymmetrical auxin response coincided with the chemotropism-exhibiting condition. We further focused on PIN-FORMED2 (PIN2), one of the PIN-family auxin efflux transporters[44], as PIN2 plays a major role in root tropisms[45] and PjPIN2 is a representative PIN localised in *P. japonicum*

root epidermal cells[19]. Using the PjPIN2-Venus module driven by the native promoter, we found an asymmetrical increase in PjPIN2 accumulation in epidermal cells of the root elongation zone towards *rac*-strigol on water agar (Fig. 3i, j, m, n), but not on a medium containing 1/2MS medium containing sucrose (Fig. 3k, l, o, p). These results suggest that local SL perception leads to local auxin accumulation for tropism, likely via PIN2, and that the nutrient-based inhibition of chemotropism occurs between SL perception and PIN2 accumulation.

## KAI2d proteins are the SL receptors involved in chemotropism
Next, we investigated if KAI2d proteins are involved in chemotropism. Using KAI2 protein sequences found in land plants, we constructed

phylogenetic trees. Seven candidate KAI2d homologues encoded in the *P. japonicum* genome[16] separated into two groups: the first group is similar to KAI2d found in obligate hemiparasites such as *S. hermonthica*, and the second group is similar to those in obligate holoparasites, such as *Orobanche* spp. (Fig. 4a and Supplementary Fig. 8). In combination with transcriptome data in *P. japonicum*[18], we found that the gene expression levels of *PjKAI2d2*, *PjKAI2d3* and *PjKAI2d3.2*, the *KAI2d* homologues most similar to those in obligate hemiparasites, were high in seedlings. In contrast, little expression was observed in the roots post infection. This pattern was consistent with the finding that the majority of *KAI2d* genes in *S. hermonthica* were highly expressed in seedlings (Fig. 4a)[29]. Next, we investigated the effects of *rac*-strigol and nutrient conditions on *KAI2* expression in *P. japonicum*. Of the 7 *PjKAI2d* genes as well as *PjKAI2i* and *PjKAI2c*, only *PjKAI2d2* and *PjKAI2d3.2* showed relatively high expression levels in basal conditions and were further increased by *rac*-strigol treatment. Basal and induced expression of these genes was attenuated in the nutrient-rich 1/2MS condition (Fig. 4b). Although nutrients also suppressed *PjKAI2d3* expression, the gene was not upregulated by *rac*-strigol, unlike *PjKAI2d2* or *PjKAI2d3.2*. The reason why *PjKAI2d3* had a different expression pattern from previous transcriptome data[18] might be because *PjKAI2d3.2* transcripts may have been mistakenly mapped to *PjKAI2d3* due to their high degree of conservation (96% amino acid identity). To test the SL-responding ability of *PjKAI2d2*, *PjKAI2d3* and *PjKAI2d3.2*, we adopted a modified cross-species complementation method, which had been successful in previous studies[27,46,47]. We used the *A. thaliana d14 kai2* double mutant in the Col-0 background[48] and evaluated responses to *rac*-strigol by seed germination rates (Fig. 4c). Since Col-0 seeds are known to lose their primary dormancy rapidly after maturation[28], we stratified seeds at 4 °C overnight after sowing to break dormancy and, therefore, to exclude the effect of dormancy. Germination phenotypes in the control *AtKAI2*-complemented lines were comparable to those in the WT Col-0 without significant changes resulting from *rac*-strigol treatment, although *D14* was still missing. Concurrently, *rac*-strigol promoted germination in *PjKAI2d2*- or *PjKAI2d3.2*-introduced lines with basal germination rates comparable to those of *d14 kai2* (Fig. 4c). These data indicate that at least PjKAI2d2 and PjKAI2d3.2 are functional as receptors of exogenous SLs.

To analyse the effects of *KAI2d* genes on chemotropism to SLs, knockout mutants obtained by gene editing would be desirable; however, *KAI2d* genes are multicopy and likely to be functionally redundant in *P. japonicum* (Supplementary Fig. 9). As we cannot yet create transgenerational *P. japonicum* transgenics, it is difficult to edit or knockdown multicopy genes[49]. Hence, we set out to generate dominant-negative plants by overexpressing the substituted KAI2d2, in which the functions of native KAI2d proteins are impeded. KAI2d proteins have a highly conserved arginine that is also conserved in D14 in *A. thaliana* and rice (Supplementary Fig. 9). Substitution of this arginine in D14 of *A. thaliana* and rice with histidine did not interfere with the hydrase activity; however, introducing the substituted D14 into each mutant did not complement the phenotypes due to loss of protein-protein interaction[50]. Therefore, we considered this substitution appropriate for generating KAI2d dominant-negative plants. We transformed *P. japonicum* seedlings with the substituted *PjKAI2d2* (*PjKAI2d2^R183H*) driven by the constitutively active promoter *PjACT*[17]. Using the resulting transgenic hairy roots, we tested chemotropism to *rac*-strigol and quantified the expression levels of *PjSMAX1.2*. Overexpression of *PjKAI2d2* promoted chemotropism to *rac*-strigol, while overexpression of *PjKAI2d2^R183H* hindered, indicating that *PjKAI2d2^R183H*-overexpressing hairy roots were dominant negative and that disturbing the interaction of KAI2d with the partner(s) was likely to cause loss of chemotropism to SLs (Fig. 4d and Supplementary Fig. 10). In addition, the SL-enhanced expression pattern of *PjSMAX1.2*, which was observed in the control plants as expected

from Fig. 2i, was diminished in the *PjKAI2d2*-overexpressing plants, whereas the expression level of *PjSMAX1.2* was constitutively enhanced in the dominant-negative plants (Fig. 4e), suggesting suppression of SL signalling. Taken together, our results indicate that PjKAI2d proteins play an important role in SL signalling, leading to chemotropism.

## Discussion

Obligate parasitic plants in the Orobanchaceae must attach to host roots within several days after germination to acquire nutrients and water because resources in their seeds are limited, and they cannot obtain sufficient energy for their survival by photosynthesis. Finding a host root to parasitise is also important for facultative parasites, which can survive without a host, as parasitisation significantly enhances its growth[14,51]. Thus, efficient growth towards host roots is critical for facultative and obligate Orobanchaceae parasites. Our study shows that both facultative *P. japonicum* and obligate *S. hermonthica* use SLs as chemoattractants, leading to host tropism (Fig. 1). As SLs do not induce chemotropism in *L. philippensis*, a non-parasitic member of the Orobanchaceae (Supplementary Fig. 2), this chemotropism phenotype may have been acquired during parasite evolution. Intriguingly, the germ tube of *O. cumana*, a holoparasite, is guided by costunolide, a sunflower-derived sesquiterpene lactone, but not by GR24, a synthetic SL analogue[8]. In comparison, *O. cumana* germination is stimulated by costunolide and GR24[8,27]. Therefore, it is likely that the nature of chemoattractants might be different among phylogenetic clades[2], in contrast with the commonality of using SLs as germination stimulants in obligate parasites. Host preferences of *Striga* spp.[3] and *Orobanche* spp.[52] may be reflected by differences in their chemoattractants, as has been supported by the findings that *O. cumana*[8], but not *S. hermonthica* (Supplementary Fig. 3), exhibits chemotropism to costunolide. It is also possible that as well as chemoattractants, host-derived quinones or phenolics designated haustorium-inducing factors, that trigger formation of haustoria in Orobanchaceae parasites[9], might be determinants of host preferences. Future identification of specific chemoattractants produced by each host may uncover the chemical basis for host preference in Orobanchaceae parasites.

SLs are known to stimulate hyphal branching of arbuscular mycorrhizal fungi (AMF)[53] that establish symbiotic relationships with plants to exchange soil-derived nutrients such as phosphate and nitrogen with plant-derived carbon sources[54,55]. Interestingly, D14L, a KAI2 homologue, is required for AMF symbiosis in rice[56]. D14L derepresses downstream signalling by removing the suppressor SMAX1, resulting in elevated AMF colonisation[40]. In our study of *P. japonicum*, we found that disturbance of KAI2d function(s) by overexpressing a substituted KAI2d2 resulted in elevated *PjSMAX1.2* expression and loss of chemotropism to SLs (Fig. 4d, e). As KAI2d-mediated SL perception for chemotropism is important in host infection to obtain nutrients, it is tempting to think that Orobanchaceae parasites may have converted the KAI2-based symbiotic AMF communication tool via MAX2 into a parasitic host-detection system. In this context, the observation that chemotropism to SLs occurs only in nitrogen-deficient conditions is intriguing (Fig. 2). Plants often produce and exude SLs when nitrogen is deficient for attracting AMF[57], which can, in turn, promote plant nitrogen acquisition. Indeed, in the case of rice, ~40% of the plant's nitrogen requirement can be acquired by AMF[58]. It is possible that, in Orobanchaceae parasites, the primary nitrogen source may have shifted from AMF to host plants, potentially by evolving KAI2-mediated signalling via MAX2[59]. Consistently, *P. japonicum* effectively transfers nitrogen from hosts, especially in nutrient-deficient conditions[60]. AMF colonisation in *P. japonicum* has not been reported, but some *Pedicularis* species, facultative hemiparasitic plants in the Orobanchaceae, can accommodate AMF[61]. How such parasites can coordinate SLs and KAI2/KAI2d proteins to accommodate both AMF and hosts is an interesting question to be answered. Future genomic

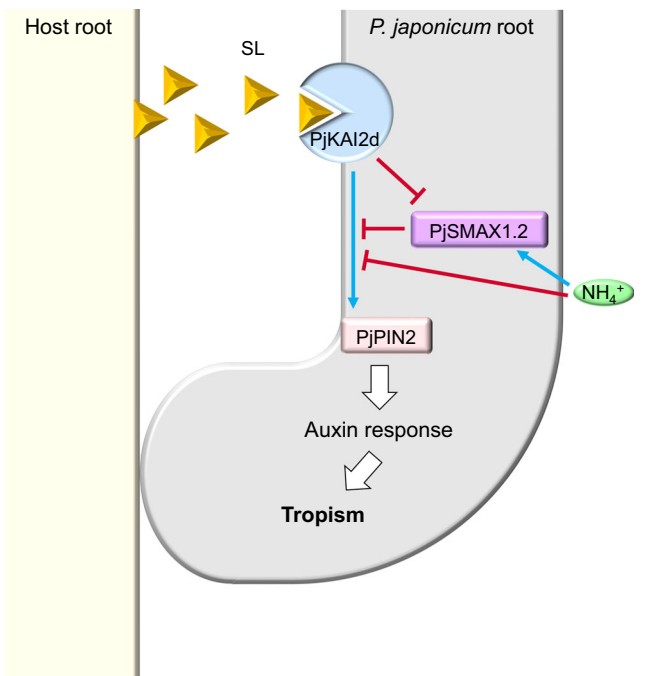

**Fig. 5 | A schematic model of tropism to host-derived SLs in *P. japonicum*.** Coloured arrows and T bars depict positive and negative regulations, respectively.

surveys of the AMF-accommodating abilities in Orobanchaceae plants may provide molecular clues of how the KAI2/KAI2d-based symbiotic-parasite relationship has evolved.

Complementation of the *d14 kai2* mutant with *PjKAI2d2* or *PjKAI2d3.2* partially rescued seed germination in an exogenous SL-dependent manner (Fig. 4c). This result indicates that SL-bound PjKAI2d2 and PjKAI2d3.2 may not efficiently bind to *A. thaliana* MAX2 and/or SMAX1. In addition, *PjKAI2d2* and *PjKAI2d3.2* failed to complement the germination-attenuated phenotype without SL application, indicating that these proteins are unlikely to bind to unidentified endogenous KAI2 ligands (KLs) to activate downstream signalling. This hypothesis is further supported by the contrast that unnatural isomers (−)-5DS and (−)-2′-*epi*-5DS activate KAI2-mediated signalling in *A. thaliana*[33], while *P. japonicum* is unresponsive to the compounds (Fig. 1h), highlighting that SLs, but not KLs, trigger chemotropism in *P. japonicum*. Similarly, *S. hermonthica KAI2c, KAI2i, KAI2d1* and *KAI2d2* do not complement the *A. thaliana kai2* mutant without SLs[27], suggesting that *S. hermonthica* may have lost KL-sensing activity for germination. Further characterisation of various KAI2 proteins and associated proteins in the Orobanchaceae family and other land plants will reveal signalling specificities differentiating chemotropism and germination functions. This research direction may eventually provide clues of how neo-functionalisation of KAI2/KAI2d occurs in plants.

Our study shows that positioning SLs beside *P. japonicum* roots induces asymmetrical accumulation of the auxin efflux transporter PIN2, resulting in the roots bending towards the chemicals (Fig. 3). This observation provides a functional link between SL and auxin signalling in the host tropism of *P. japonicum* (as modelled in Fig. 5). To date, the effects of SLs on auxin-mediated processes have been studied primarily in *A. thaliana*[62,63]. SLs interfere with auxin-mediated PIN polarisation in *A. thaliana* in a MAX2-dependent manner by derepressing PIN endocytosis, resulting in the interference of auxin canalisation[62]. In roots, asymmetrical PIN2 accumulation in epidermal cells is a key factor in root bending, as seen in gravitropism[64]. In *P. japonicum*, exogenous SLs are perceived by the epidermal cells in the elongation zone, as indicated by YLG-based fluorescence, where PjPIN2 accumulates (Figs. 1i and 3n). This result is consistent with the finding that KAI2

is required for the local accumulation of PIN2 in *A. thaliana* roots[65]. As *A. thaliana* does not exhibit chemotropism to SLs (Supplementary Fig. 2c, d) and frequently SLs are evenly distributed throughout *A. thaliana* media, it is difficult to compare the effects of SLs on PIN2 accumulation in *A. thaliana* and *P. japonicum*. An alteration in SL-PIN2 relationships may have occurred in Orobanchaceae parasites that enabled chemotropism. Importantly, SLs can be perceived in nutrient-rich conditions (Fig. 2e), but the asymmetrical PIN2 accumulation occurs only in nutrient-deficient conditions (Fig. 3n, p). Given that chemotropism to SLs were halted by exogenous nitrogen sources, especially ammonium ions (Fig. 2 and Supplementary Fig. 5), asymmetrical PIN2 accumulation may also be regulated by nitrogen levels. This is reasonable because *PIN2* expression and the subsequent asymmetric auxin distributions resulting in root gravitropism are inhibited by ammonium ions in *Arabidopsis*[66–68]. The SL signalling pathway is also affected by ammonium ions, as indicated by upregulation of *SMAX1.1* and *SMAX1.2* (Fig. 2i–k). This finding suggests that ammonium ions regulate root tropism by controlling PIN2 accumulation potentially via SMAX regulation (Fig. 5). As rice roots exhibit chemotropism to ammonium ions[69], it would be interesting to test whether that tropism to ammonium ions also involves an asymmetric accumulation of PIN2, which may be regulated by SL signalling in rice.

In summary, our study unveils important molecular clues about host tropism in Orobanchaceae parasitic plants, providing a function of SLs as chemoattractants. We expect this study will encourage additional investigations to elucidate tropism processes, one of the important steps for infection in parasitic plants. Such future studies may encourage design solutions for protecting agricultural fields from nuisance weeds.

## Methods
### Plant materials and growth conditions
Unless otherwise noted, *P. japonicum* (Thunb.) Kanitz seeds and *A. thaliana* seeds (Col-0, *d14 kai2*[48] and complemented lines) were germinated on 1/2MS medium (0.8% (w/v) INA agar, pH 5.8) containing 1% (w/v) sucrose. Before sowing, seeds were sterilised with a diluted commercial bleach solution (Kao, Tokyo, Japan, 10% (v/v) for *P. japonicum* and 5% (v/v) for *A. thaliana*) for 5 min and rinsed at least five times with sterilised water. Plate-sown seeds were stratified at 4 °C in the dark for 1 to 3 night(s), then grown horizontally in long-day conditions (16-h light (-40 and ~30 µmol m⁻² s⁻¹ for *P. japonicum* and *A. thaliana*, respectively), 8-h dark at 25 °C and 22 °C for *P. japonicum* and *A. thaliana*, respectively). *P. japonicum* and *A. thaliana* seeds were grown at 70% and 50% humidity, respectively, to minimise deviation of results due to humidity. *S. hermonthica* (Del.) Benth. seeds were carefully handled with glass pipettes as the seeds stick to plastic pipette tips. Seeds were soaked in a diluted 20% (v/v) commercial bleach solution during a brief vortex, then sterilised with a fresh bleach solution for 5 min, followed by at least five rinses with sterilised water. Seeds were then soaked in 5-ml of sterilised water in 6-well plates and incubated in the dark at 25 °C for 1 to 2 weeks. To induce germination, water was replaced with a 10 nM (+)-strigol[70] solution, and the plates were incubated for 4 h in the dark at 25 °C. The (+)-strigol solution was then replaced with water, and the plates were incubated at least 24 h before further analysis to exclude the residual effects of (+)-strigol. *L. philippensis* (Cham. & Schltdl.) Benth. seeds were sterilised with a diluted commercial bleach solution (Kao, Tokyo, Japan, 5% (v/v)) for 5 min and rinsed at least five times with sterilised water before soaking in 5-ml of sterilised water in 6-well plates. Water-soaked seeds were grown in long-day conditions (-30 µmol m⁻² s⁻¹) at 22 °C for 6 days. For further analyses, seedlings were carefully handled with glass pipettes and a pair of tweezers. *O. sativa* (japonica, c.v. Shiokari) seeds of WT and *d10*[35] were sterilised with 70% (v/v) ethanol for 3 min, followed by a diluted commercial bleach solution (Kao, Tokyo, Japan, 50% (v/v)) for 30 min and rinsed with sterilised water at least five times. Surface-

sterilised seeds were then grown vertically on 0.6% (w/v) water agar in long-day conditions (18 h /6 h light/dark, ~120 μmol m$^{-2}$ s$^{-1}$) at 25 °C.

## Chemicals

SLs and analogues used in this study are listed in Supplementary Data 1. Chemicals were stored at −20 °C as 100 mM (YLG) or 10 mM (the other chemicals) stocks in DMSO.

## Chemotropism assays

For assays using *P. japonicum and A. thaliana*, seedlings (3-day-old for *P. japonicum* and or 2-day-old for *A. thaliana*) were transferred carefully to solid media containing nutrients (0.8% (w/v) INA agar, pH 5.8), solid media containing ammonium ions at a neutral condition (0.8% (w/v) INA agar, pH 7), or solid media without nutrients (0.7% (w/v) INA agar, pH 5.8), incubated vertically for 1 day, and transferred to the media used for the assays. Note that because of the difference in the ionic concentration, we made the difference of INA agar concentrations between media to approximately match hardness. For assays using *S. hermonthica* and *L. philippensis*, germinated seedlings were transferred carefully to solid medium without nutrients (0.7% INA agar (w/v), pH 5.8). *P. japonicum* transgenic hairy roots were generated from seedlings 3- to 4-weeks post-transformation and were transferred carefully to solid medium without nutrients (0.7% (w/v) INA agar, pH 5.8), incubated horizontally for 2 days, and transferred to new solid medium without nutrients. Filter paper disks (4-mm diameter) were soaked with the chemical-containing solution before being placed 5-mm (for *P. japonicum* and *A. thaliana*) or 3-mm (for *S. hermonthica* and *L. philippensis*) from the seedlings. We defined seedlings that exhibited chemotropism as showing root bending of 30° or more towards a filter paper disk containing a tested chemical.

## Microscopy

For fluorescent stereo microscopy, photos were taken using a fluorescence stereo microscope (M165 FC, Leica) before and after chemical-treated seedlings were incubated in the dark at 70% humidity for 1 day, 25 °C for *P. japonicum* and *S. hermonthica*, and 22 °C for *A. thaliana* and *L. philippensis*, respectively, except for time-lapse microscopy. For time-lapse microscopy, 100 photos were taken throughout 18 h using a fluorescence stereo microscope (M205 FA, Leica) in the dark at 25 °C. For confocal microscopy, photos were taken using an inverted confocal microscope (TCS SP5 II, Leica). Fluorescein from hydrolysed YLG was excited with a 488-nm laser, and the detected emission spectra were observed at 500–531 nm. Excitation and detection of Venus and mCherry fluorescence were performed as described previously[12]. The bending degree and image analyses were measured using ImageJ[71].

## Host chemotropism assay for *P. japonicum*

*P. japonicum* seeds were sterilised, sown, stratified, and grown on solid growth medium containing 1/2 MS, 0.6% (w/v) agar, and 1% sucrose (pH 5.8) for 3 days in the dark at 25 °C. The seedlings were then transferred to 0.6% (w/v) agar medium without nutrients and grown vertically for 1 day in long-day conditions at 25 °C. Rice seeds were sterilised, sown, and grown for 4 days as described in the "Plant materials and growth conditions" section. Host chemotropism assays were then performed on 0.6% agar medium without nutrients in long-day conditions at 25 °C as follows: a pair of WT rice and *d10* rice seedlings were transferred to agar medium in a square petri dish (140 mm × 100 mm × 15 mm). Single roots of similar length from each genotype were chosen and aligned vertically in parallel at a distance of ~2 mm. A *P. japonicum* seedling was then placed between two rice roots and a single *P. japonicum* root was carefully placed parallel to the middle of the rice roots. Relatively young root regions (~500 mm from the tip) of the host were chosen for the initial infection position (0 days) because these areas produce few to no lateral roots during the observation period that would otherwise physically disturb *P. japonicum* roots from bending. Three

independent experiments were conducted with each replicate containing 7 pairs. For about half of the pairs, WT seedlings were placed at the left and *d10* seedlings at the right, and for the rest vice versa. To keep the roots on the medium, a cover glass was placed on the top of the roots. The plates were sealed with surgical tape and positioned vertically in the growth chamber. Images were captured 0, 1, and 2 day(s) after infection using a wide zoom stereo microscope (Olympus SZX16).

ImageJ (version 1.52q), Microsoft Excel and Adobe Illustrator software were employed to quantify the images. In all cases, WT roots were positioned on the left and *d10* was on the right. First, images derived from 1- or 2-day(s) post infection were superimposed onto their corresponding 0-day images using Adobe Illustrator. Next, using ImageJ, the position of the *P. japonicum* root tip was defined as $y = 0$ and the position of the left (WT) host at horizontal axis with the *P. japonicum* root tip was defined as $x = 0$ in 0-day samples. Thus, each rice root and the *P. japonicum* root tip were assigned a unique $x$ and $y$ value: for instance, WT rice ($x = 0$ μm, $y = 0$ μm), *P. japonicum* ($x = 904.8$ μm, $y = 0$ μm) and *d10* rice ($x = 1979.6$ μm, $y = 0$ μm) at 0 day. The $xy$ positions of the *P. japonicum* root tip at 1- and 2-day(s) post infection relative to 0 day were then measured. Raw $xy$ data generated from all samples as described above were subjected to centring and scaling processes before generating the graphs shown in the figures (For details refer to Source Data).

## Extraction of total RNA and RT-qPCR

To extract total RNA from *P. japonicum* seedlings, 3-day-old seedlings were transferred to media containing varying nutrients and were grown vertically for 1 day. The seedlings were transferred again to the same medium used for the assays. At 3 h after incubation, root tips were excised and immediately frozen in liquid nitrogen. To extract total RNA from *P. japonicum* transgenic hairy roots, root tips of the generated hairy roots were excised and immediately frozen in liquid nitrogen after the chemotropism assays (see "Chemotropism assays" section). Total RNA extraction, cDNA synthesis and RT-qPCR were performed as previously described[12]. In brief, we used the RNeasy plant mini kit (QIAGEN) to extract total RNA. To remove residual genomic DNA, DNase treatment was performed during RNA extraction. The resulting total RNA was used for cDNA synthesis with ReverTra Ace qPCR RT Kit (TOYOBO). RT-qPCR was performed using Thunderbird SYBR Green and ROX (Toyobo) and a real time thermal cycler mx3000p (Stratagene). We used *Polyubiquitin C 2* (*UBC2*) as a reference gene to normalise the expression levels. The expression level of each gene was quantified using the ddCt method (dd, delta-delta). Primer sequences are provided in Supplementary Data 2.

## Cloning

The constructs *pDR5::3xmCherry-NLS* and *pPjPIN2::PjPIN2-Venus*[19] were previously described. pKAI2pro-GW[72] was used as the Gateway destination vector to express each protein under the control of the *AtKAI2* promoter. The AtKAI2-expressing vector pKAI2-AtKAI2 was previously generated[72]. PjKAI2d-expressing vectors were constructed as follows: the genomic regions containing each coding sequence (CDS) of *PjKAI2d* was PCR amplified. With the resulting PCR products, each PjKAI2d CDS was PCR amplified with primers containing attB sites, subcloned into pDONR/Zeo (Thermo Fisher Scientific) by Gateway BP cloning (Thermo Fisher Scientific) to generate entry vectors, then cloned into pKAI2pro-GW by Gateway LR cloning (Thermo Fisher Scientific), yielding pKAI2-PjKAI2d2, pKAI2-PjKAI2d3 and pKAI2-PjKAI2d3.2. For constructing the expression vectors to transform *P. japonicum*, we used GoldenGate technology for assembling modules[73]. The GoldenGate modules containing the *35S* promoter (*p35S*, pICH51266) or *35S* terminator (*35St*, pICH41414) were pre-existing[73]. Modules containing the actin promoter (*pPjACT*), 3xVenus-NLS CDS or HSP18.2 terminator with the 3′-untranslated region (*HSPt*) were

described previously[13,17]. A single mutation was introduced into the *PjKAI2d2* CDS using a KOD plus Mutagenesis kit (TOYOBO) with the entry vector described above as a template. The native *PjKAI2d2* CDS, the mutated *PjKAI2d2* CDS and the *PjSMAX1.2* CDS were PCR amplified and cloned into pICH41308 to generate level 0 CDS1 modules, respectively. For removal of the *Bsa*I and *Bpi*I sites in the *PjSMAX1.2* CDS, a part of the CDS (2347-3864) was de novo synthesised with the *Bsa*I and *Bpi*I sites synonymously substituted. The rest part of the level 0 CDS1 module containing *PjSMAX1.2* CDS was amplified by inverse PCR. The two fragments were fused using In-Fusion® HD Cloning Kit (TAKARA), yielding a level 0 CDS1 module with the *Bsa*I and *Bpi*I sites in the *PjSMAX1.2* CDS mutated. The modules were assembled and cloned into level 1 vectors to generate *pPjACT::PjKAI2d2:HSPt*, *pPjACT::PjKAI2d2^{R183H}:HSPt*, *pPjACT::PjSMAX1.2:HSPt* and *p3SS::3xvenus-NLS:3SSt* transcription units. The resulting transcription units were further combined and cloned into the level 2 binary vector pAGM4723. End-linkers and dummy modules were used as needed. Primer sequences are provided in Supplementary Data 2

## Transformation of *P. japonicum*

*Agrobacterium rhizogenes* AR1193 strain was used to transform *P. japonicum* seedlings. *A. rhizogenes*-mediated transgenic hairy roots were generated and identified according to previously described methods[12,13]. In brief, just prior to transformation, Silwet L-77 (Bio Medical Science) was added to an *A. rhizogenes* bacterial suspension (OD600 = 0.1) to a final concentration of 0.02% (v/v), followed by gentle mixing by inversion. Six to seven millilitres of the bacterial suspension were transferred to 15-ml plastic tubes. Six-day-old *P. japonicum* seedlings were soaked in the bacterial/Silwet L-77 suspension. The seedlings were subjected to ultrasonication using a bath sonicator (Ultrasonic automatic washer, AS ONE, Japan) at room temperature for 10 to 15 s. The sonicated seedlings and the bacterial/Silwet L-77 suspension were transferred to Petri dishes containing a filter paper. The dishes were sealed with surgical tape and subjected to a 5-min continuous vacuum infiltration. For co-cultivation, the seedlings were transferred to freshly made Gamborg B5 plates (0.8% (w/v) INA agar, pH 5.8) containing 1% (w/v) sucrose and 450 μM acetosyringone, followed by incubation in the dark at 22 °C for 2 days. The seedlings were then transferred to Gamborg B5 plates (0.8% (w/v) INA agar, pH 5.8) containing 1% (w/v) sucrose and 300 μg ml⁻¹ cefotaxime and grown horizontally in long-day conditions for 3 to 4 weeks as described in the "Plant materials and growth conditions" section. The resulting transformed roots were used for the assays. We identified the transformed roots by detecting the fluorescence from the marker protein using a fluorescence stereo microscope (M165 FC, Leica).

## Phylogenetic analyses

We used the CLC Main Workbench (ver. 8.0, Qiagen) for phylogenetic analyses. The CDSs of KAI2 used for the phylogenetic analyses in Conn et al.[27] were automatically translated and used in our study. Two candidate genes, KAI2d3.2 and KAI2d4.2 in *P. japonicum*, which were newly identified from the *P. japonicum* genome by BLASTp analysis using PjKAI2d1-PjKAI2d5 as queries (with e-values under 1e⁻¹⁰⁰), were translated to amino acid sequences and added to the KAI2 sequence group. We also added amino acid sequences of KAI2 homologues in *Phelipanche ramosa*[47] to the KAI2 sequence group. The KAI2 sequences processed by trimAL v1.2[74] using the automated1 settings were aligned using default settings. After alignment, phylogenetic trees were drawn using the maximum-likelihood method with 1000 bootstrap repetitions. We generated the figures using iTOL v6 (https://itol.embl.de/[75]).

## Transformation of *A. thaliana* and cross-species complementation assays

pKAI2-AtKAI2, pKAI2-PjKAI2d2, pKAI2-PjKAI2d3 or pKAI2-PjKAI2d3.2 were electroporated into *Agrobacterium tumefaciens* GV3101

competent cells. The resulting bacterial cells were used to inoculate flowers in a transformation method modified from Martinez-Trujillo et al.[76]. For germination assays, *A. thaliana* T2 seeds that were at least 1-month old were used. Seeds were sterilised with chlorine gas for 2–3 h in a safety cabinet and dried for at least 2 h in a clean bench, followed by sowing on solid media without nutrients (0.7% (w/v) INA agar, pH 5.8) containing 1 μM *rac*-strigol or 0.1% (v/v) DMSO. Plate-sown seeds were stratified at 4 °C in the dark for 1 night, then incubated horizontally in the dark. Germination rates, defined by radicle emergence, were scored periodically.

## Statistical analyses

Welch's *t* test and two-way ANOVA, Tukey's multiple comparison test were performed in Microsoft Excel 2016 and GraphPad Prism version 8.1, respectively. Details of statistical analyses including statistical methods, numbers of individual batches, numbers of technical replicates, numbers of biological replicates, and statistical significances are described in each figure legend.

## Reporting summary

Further information on research design is available in the Nature Research Reporting Summary linked to this article.

## Data availability

Transcriptome data for *S. hermonthica*[29] and *P. japonicum*[18] are available from the DNA Data Bank of Japan (http://www.ddbj.nig.ac.jp/) under accession numbers DRA008615 and DRA003608 for *S. hermonthica* and DRA010010 for *P. japonicum*, respectively. Sequence data from this study have been deposited at the GenBank/EMBL libraries. Accession numbers are provided in Supplementary Data 3. Gene IDs of the *P. japonicum* genes investigated in this study are provided in Supplementary Data 3. Source data are provided with this paper.

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

## Acknowledgements

We thank Prof. Abdel G. Babiker (Environment and Natural Resources and Desertification Research Institute, Sudan), Prof. Yuichiro Tsuchiya (Institute of Transformative Bio-Molecules, Nagoya University), Prof. Lam-Son Phan Tran (Institute of Genomics for Crop Abiotic Stress Tolerance, Texas Tech University) and the late Prof. Kenji Mori for sharing *S. hermonthica* seeds, *L. philippensis* seeds, *A. thaliana d14 kai2* mutant seeds and (+)-strigol, respectively. This work was supported by Ministry of Education, Culture, Sports, Science and Technology KAKENHI grants (19K16169 to S.C., 20H05909 and 21H02506 to K.S. and S.Y., 17H06172 and 22H00364 to K.S.); the Japan Society for the Promotion of Science (JSPS) KAKENHI Grants-in-Aid for JSPS Fellows (JP21J00718 to S.O.); JST PRESTO (JPMJPR194D to S.Y.).

## Author contributions

S.O., D.C.N., S.Y., and K.S. conceived and designed the study; K.S. supervised the experiments; S.O., S.C., and A.R.F.W. conducted the experiments; all authors analysed the data; S.O. and K.S. drafted the original manuscript; all authors critically revised the manuscript and approved the final version.

## Competing interests

The authors declare no competing interests.
