## [Peer Review File · Nature Communications]

Strigolactones are chemoattractants for host tropism in
Orobanchaceae parasitic plantsREVIEWER COMMENTS

Reviewer #1 (Remarks to the Author):

The Authors studied the novel chemoattractive feature of strigolactones (SLs) in plants. Although SL-induced chemotropism in fungi is a more or less explored phenomenon, this present paper is the very first genuine report in plants. It is presented in an excellent way that natural SLs, unnatural analogues and agonists trigger chemotropism in two hemiparasitic Orobanchaceae genus. I would like to especially praise the use of rice d10 (MAX4) mutant to demonstrate SL-dependent chemotropism in the presence of host plant. The Authors also found that non-starving *P. japonicum* seedlings, non-parasitic Orobanchaceae, as well as *Arabidopsis* are not responsive to SLs in this context, therefore they concluded that this function of SLs is restricted to nutrient-starving hemiparasites. The Authors also found evidence that chemotropism in *P. japonicum* is mediated by auxin responses and the divergent members of KAI2d clade.

My overall impression is that the manuscript is generally well written and the literature of this field has been cited appropriately and in the right context. The experiments have been performed accurately with an adequate number of replicates and evaluated with the proper statistical method. The „Results“ and „Discussion“ section, as well as the „Methods“ are presented in a clear fashion. Supplementary materials, including the videos are all OK. I can't see any flaws regarding the methodology, experimentation and data interpretation presented in this paper.

I have a few suggestions and some comments on the MS. These are:

-line 91: the unnatural enantiomer (-)-5DS (or ent-5DS) signals through the „canonical“ KAI2 in *Arabidopsis* (Scaffidi et al., 2014). *P. japonicum* is unresponsive to this compound, highlighting that SLs and not KAI2-ligands trigger chemotropism. I suggest that the Authors should comment on this in the text.

-line 116 and onward: Authors cannot directly compare chemotropism data obtained from water-agar and 1/2MS+sucrose plates, because two factors were changed simultaneously. Later it became clear that macronutrients obviously inhibit chemotropism in the presence of sucrose, hence, mineral nutrients and not the carbon source/sugar signaling is responsible for the halting chemotropism. I suggest that the Authors should rewrite these sentences to make it clear that the conclusion were not drawn from the direct comparison.

-Figure 2h and k: when ionic forms are referred in the text, please use NH₄⁺ and NO₃⁻.

-line 132: downstream SL signaling has been investigated with monitoring PjSMAX1. In *Arabidopsis*, SMAX1 expression is primarily regulated through KAI2. I understand that SL-specific transcriptional changes are weak in most cases and one cannot expect spectacular fold changes upon SL treatment. Why did the authors use SMAX1 and wonder whether they gave a try with other reported SL-specific genes from dicots such as SMXL7, TCP1 or PAP1 (Wang et al., 2019), or something else?

-line 151 and onward: again, my concerns are more or less the same as in line 116. I suggest to

-1. include 1/2MS agar and water-agar+sucrose data, or

-2. omit 1/2MS+sucrose data from the figure and move it to a supplementary figure (this half of the figure does not add too much for the context, I think), or

-3. extensively explain in the text that not sugar, but the micronutrient/macronutrient constitution of the plates affects chemotropism.

-line 193 and onward: real-time PCR data of the PjKAI2 dominant negative mutant expression level should be included, although I do not stick to it. Also, on Figure 4, a photograph of a non-responding transformant should be included (in a similar fashion as on Figure 1.).

-line 183: please change PjKAI2d to PjKAI2d2.

-line 288: sources of the *P. japonicum*, *L. philippensis* and *S. hermonthica* seed batches should be presented.

-line 318 and onwards: the solid media with nutrients contained 0.8% (w/v) agar, while the media without nutrients contains 0.7% (w/v) agar. What is the reason for this?

-Also in this section, the following sentence has been repeated four times, with only the species name

changed: "germinated seedlings were transferred carefully to solid medium without nutrients (0.7% INA agar (w/v), pH 5.8). Filter paper disks (4-mm diameter) were soaked with a chemical-containing solution, before being placed 3-mm from the seedlings.". Please use only one sentence with listing all the species on which the method has been applied. E.g.: "For assays using *P. japonicum*, *S. hermonthica*, *L. philippensis* and *Arabidopsis*, germinated seedlings were transferred...and so on.

-line 341: please use "fluorescent stereo microscopy" instead of "fluorescence".

-line 412: "dummies" means oligonucleotide linkers/modules? Please clarify.

Reviewer #2 (Remarks to the Author):

The MS by Ogawa et al. describes the interesting results: 1) Chemotropism of *P. japonicum* to SLs; 2) Inhibition of chemotropism to SLs by ammonium; 3) Correlation between chemotropism and auxin. However, solid evidences in the MS are not enough to clarify the underlying mechanism regulating chemotropism. In reaching the required standard I feel that there is a serious omission in the experimental data provided.

Major comments:

- 1) Genetic evidences are really required to support the connection between tropism and ammonium. What's about the chemotropism of PjSMAX1.2 loss- and gain-of-function lines in response to ammonium ions?
- 2) Genetic evidences are also required to support the connection between tropism and auxin. What's about the tropism of pin loss- and gain-of-function lines in response to ammonium ions? And, what's auxin and PIN distribution in dominant-negative plants PjKAI2d2R183H and PjSMAX1.2 loss-/gain-of-function lines?
- 3) Chemotropism is related with ammonium and auxin. In order to draw the pathway of SLs regulated chemotropism, authors should clarify the connection between ammonium and auxin. Again, genetic evidences should be provided.
- 4) In *Arabidopsis*, PIN2 and PIN3 are important for gravitropism. What about PIN2 trafficking and PIN3 polarization during chemotropic response of *P. japonicum*?
- 5) Ammonium uptake could induce acidic stress. Does different pH affect chemotropism? Ammonium regulated chemotropism is dependent on pH?
- 6) Exogenous SLs are perceived as indicated by YLG-based fluorescence. It's good to also directly measure endogenous strigolactone contents in roots of *P. japonicum*.
- 7) Regarding to the biological significance, it's better to have field test to compare the effect of *P. japonicum* to the host plants under abundant and limited nutrient conditions.

Minor comments:

- 1) Proposed model illustrating SLs acting as navigators for parasite roots should be provided.
- 2) The statistical test (t-test) is not appropriate for multiple comparisons in some graphs. ANOVA analysis should be applied.

Reviewer #3 (Remarks to the Author):

The authors have made a remarkable discovery of chemo-attraction of weed roots towards strigolactones (SLs) exuded by nearby host roots. The SL-response is detected by KAI2d SL receptors, associated with auxin asymmetry and inhibited by ammonium. Unlike some other parasitic weeds, *Phtheirospermum japonicum* is a parasitic weed that does not require SLs to trigger germination near

a host root, and does not necessarily need a host. It also contains SL receptor genes. Evident question are; is this a real biological phenomenon, how does a weed know when a host is nearby so that it can attack, what is the mechanism of the tropism and does that mechanism involve SLs and auxin? A broader implication is that by understanding such a mechanism, we might be able to figure out how to exploit that knowledge to disrupt weed attack and save crops.

The authors acquired the following results:

Weed roots showed dramatic growth towards a range of natural SL compounds in soaked filter paper disks.

Weed root growth towards host root was not as good a rice mutant that fails to produce SLs.

Ammonium suppressed weed root growth towards the host, perhaps suggesting that the weed may sense ammonium as a signal that host colonisation is not beneficial for survival in nutrient rich soil.

Increase in auxin reporter and asymmetrical increase in YLG and PjPIN2 reporter proteins suggest that the signal differentially affects one side of the root, and auxin causes the bending towards SLs. (Auxin and PIN2 are well-known root tropism regulators.)

PjKAI2d2 and PjKAI2d3.2 showed relatively high expression in basal conditions and were further increased by SL treatment, making them candidates for perception in weed root cells of host SLs.

PjKAI2d and PjKAI2d3.2 were found to be functional as SL receptors when transformed into Arabidopsis and tested for germination stimulation by exogenous SLs.

It was not feasible to create transgenerational japonicum transgenics of edit or knockdown multicopy genes. Therefore, a dominant negative version of PjKAI2d2, PjKAI2d2-R183H, driven by a constitutively active promoter was made, which would be expected to interfere with and counteract endogenous KAI2d function. It disrupted weed chemotropism, suggesting that KAI2d SL receptors in weeds facilitate chemotropism response. PjKAI2d2-R183H also caused feedback on the endogenous weed SL pathway, suggesting the construct had a repressive function.

The study is convincing. Is it novel? The authors note the recent study that *Orobanche cumana* respond to costunolide for chemo-attraction (Krupp et al., 2021). Costunolide looks chemically similar SLs or karrikins. So it is important for the field to know that *Phtheirospermum japonicum* is responding to (natural) SLs rather than costunolide. In future, it would be good to see if *Phtheirospermum japonicum* responds to other compounds like costunolide (did the authors test that?) and to see how extensive the diversification in weed species is, for the chemo-attraction compound response.

In future, it would also be fascinating to figure out how SL-related compounds and KAI2d transmit the signal to promote auxin asymmetry in weed roots, and thus causes root bending. As the authors mentioned, other studies have shown that SLs (and karrikins) can repress auxin transport and interfere with the auxin-inhibition of PIN protein endocytosis, which does not require de novo protein production, etc. However, the cellular mechanism remains unknown. There seems to be a specific effect of AtKAI2 on gravitropism and PIN2 (Swarbreck et al., Plant Journal, 2019; Villaecija-Aguilar et al., PLOS Genetics, 2019; Villaécija-Aguilar et al., Current Biology, 2022).

Suggested edits

L85 "rac-strigol (STR)" Sometime too many abbreviations make things harder to read. And I've never seen it written STR before. Would it be simpler if you just write rac-strigol? It is not a long word. (It's ok to keep it in the figures.)

L111 "SLs are essential for host tropism". Could you please clarify this statement in the manuscript? Could there be other signals, but SLs override them?

L143 & 149 Do you want to state that the asymmetry was towards the attractant, here and elsewhere?

L230 "it is tempting to think". Could the mechanism have evolved in different ways related to nitrogen-deficient conditions? For instance, there is growing literature on the specific root effects of ammonium. This is a recent review on the topic (Jia and von Wirén, J Exp Bot., 2020). Ammonium has been observed to disrupt root growth and gravitropism tropism in non-weed species, and associated with inhibition of PIN auxin transport. It has been suggested that ammonium can be somehow toxic to roots, but I suspect that ammonium acts as a signal to switch from root foraging, in nutrient poor soils, to root slowing and proliferating in a nutrient rich soil zone. Is it possible that SLs/karrikins/KAI2-ligand and ammonium converge on the same cellular mechanism of auxin transport inhibition, and that the *Phtheirospermum japonicum* lineage has co-opted this endogenous SLs/karrikins/KAI2-ligand and ammonium root response mechanism to enhance host interactions? The supplementary video suggests that the weed root actually slows down and thickens upon touching a potential host root, which would fit with SLs acting to repress auxin transport? Something similar can happen with NPA-treated *Arabidopsis* roots. Perhaps this is a simpler alternative concept, particularly if AMF is not present in *Phtheirospermum japonicum*? However, I agree that a problem with this idea is that SL-, KL-like signals, NPA or ammonium, do not appear to induce chemo-attraction in *Arabidopsis*? Does this mean there is some additional mechanism in the weed to make the epidermis PIN2 hyper-sensitive? Would the authors like try something like NPA or ammonium in a filter disk?

Reviewer #4 (Remarks to the Author):

The ability to recognize a host plant is crucial for obligate parasitic plants and benefit for the facultative forms. This recognition implies allelochemicals including germination stimulants (often strigolactones for obligate forms), chemoattractant compounds directing the parasite's growth towards its host and compounds inducing the formation of the haustorium (HIFs), the organ that binds the parasite to its host. So far, while much work has focused on identifying germination stimulants and HIFs and deciphering induced signalling, the existence of host chemotropism, via strigolactones, was strongly suspected but still to be demonstrated, what Ogawa and his collaborators have achieved in this manuscript. In addition, the results presented here also lead to an interesting discussion by the authors concerning (1) the implication of this chemotropism in the preference of hosts in parasitic plants, (2) the diversion by parasitic plants of this perception system originally active in plants to promote mycorrhizal symbiosis, and thus (3) the potential place of mychorization, in addition to host plants, in the mineral nutrition of parasitic plants. In this context, the results presented by Ogawa and his collaborators represent a major advance in the understanding of plant-plant interactions of a parasitic order and draw very interesting research perspectives.

Thus, the results presented in this manuscript demonstrate (1) the positive chemotropism of exogenous strigolactones (natural and synthetic) and of the host (rice) towards the growth of the root (*Pterospermum*) or the radicle (*Striga*) of the parasitic plant; (2) the asymmetrical perception of strigolactones by host root epidermal cells; (3) the involvement in *Pterospermum* of KAI2d proteins (KAI2d2, KAI2d3.2) in strigol perception, leading to asymmetrical overexpression of PIN2 (auxin efflux transporter) in roots. Some results also suggest the specificity of strigolactone chemotropism in parasitic Orobanchaceae since no strigol chemotropism has been detected in a parasitic Orobanchaceae (*Lindenbergia philippensis*) and *Arabidopsis* (autotrophic plant). The inhibitory effect of nitrogen nutrients (ammonium especially) on the chemotropism of strigol towards *Pterospermum* roots is also demonstrated. This effect expresses in post perception of strigol, with the overexpression of SMAX1, a negative regulator of strigolactone signalling.

For this, the authors have developed a chemotropism test of chemical compounds and host plants towards young plants of *Pterospermum* (facultative parasite) and germinated seeds of *Striga hermonthica* (obligate parasite). The use of the fluorescent synthetic strigolactone (YLG) has made it possible to visualize the perception of strigolactones by root epidermic cells and thus to discern the effect of chemotropism inhibitors (nitrogen) on receptors (KAI2d) or induced signalling. The involvement of strigolactones and KAI2d receptors in host chemotropism has been validated by using

negative non strigolactone d10 rice mutants and dominant *Pterospermum* hairy roots over-expressing a mutated KAI2d protein.

The work and the manuscript are of high quality. Nevertheless, some questions remain. Some points deserve clarification to improve the manuscript before publication. My remarks and questions are listed below:

Introduction

P3L8. Reference 7 is of historical importance but seems inappropriate here to illustrate the relatively widespread involvement of strigolactones in the germination of obligate root-parasitic plants.

P3L11 (end of paragraph). I agree, but I think it is useful to quote, here in this introduction part, the previous work on the positive chemotropism of lactone sesquiterpenes towards the obligate parasite *Orobanche cumana*, because of its pioneering character.

P4L13. Ref 27-29. Need to include the work recently published in *Plant Comm* on PrKAI2d in the obligate parasite *Phelipanche ramosa* (De Saint Germain et al. 2021 in *Plant Comm.*). Integration of these data into the KAI2 phylogenetic tree proposed by the authors should be also required (updating of the data).

Results

P5L15-16. Please specify the reasons for choosing these three natural strigolactones for the study. This is likely useful for people who are not specialists in parasitic plants.

P6L3. 100 nM (+)-5DS induces a significantly different tropism on figures 1g (about 50%) and 1h (only 20%). Thus, for the same treatment, the results differ from one set of experiments to another. How do you explain this? Does this question the reliability and robustness of the chemotropism test and therefore the results?

P6L2-3. The chemoattractivity of (+)-5DS was comparable to that of (+)-2'-epi-5DS: indeed at 1 μ M, but not at 100 nM. So, is the activity of these two SLs really comparable?

P6L7 (Fig. 1i): YMG 100 μ M for Fig. 1i. Is it correct? (much more concentrated than for tropism tests (Fig. 1f,g).

P6L8-9 "indicating that asymmetrical SL recognition activity induces an asymmetrical root elongation pattern, leading to chemotropism." Here, "indicating" seems too affirmative because the authors do not demonstrate the causal link between asymmetric recognition of chemoattractant and chemotropism. This requires correction.

P6 L11-13. "We found that *S. hermonthica* exhibited chemotropism to rac-STR and had an asymmetrical YLG recognition pattern like *P. japonicum*". According to Fig.1m, this statement seems abusive. Indeed, the asymmetrical pattern of YMG recognition is sincerely not evident in the experiment with *Striga* germinated seeds. Please, comment this major result concerning the asymmetric status of strigolactones perception, which is assumed by the authors.

P6 L14-18 Additional results, notably with a wider panel of strigolactones are needed to assume the idea that chemotropism of strigolactones would be specific to parasitic plants. Are these data available? (supplementary data?)

P7L13-14 (Fig. 2d). In contrast to macronutrients (+ sucrose), micronutrients are tested without sucrose, why? This requires clarification.

P8L6-8 (Fig. 2j,k). Data on the effect of nutrients (Fig. 2j,k) on PjSMAX1 expression are missing. Why?

P9L14. It would be appreciated if Fig.4a was updated with recent KAI2d data in *P. ramosa* (De Saint Germain et al. 2021 in *Plant Comm.*).

P10L5-10. "from previous transcriptome data". Please indicate references. Indeed, the discrepancy in results between previous and present analyses is regrettable. The explanation given by the authors is plausible. Is a verification possible, has it been done?

P10L17. PrKAI2d2 instead of PrKAI2d

P11L16 (Fig.4e). SMAX1.2 expression in control plants increases in response to strigol. Is an expected result? Comments are required.

Discussion

P12L6. Is the reference 6 appropriate?

P12L8-10. Are KAI2 data available for *L. philippensis*? if so, can they support the author's hypothesis?

P12L10-15. Has the chemiotropic effect of costunolide been tested by the authors against *Pterospermum* and *striga*? These tests should be necessary to support that susceptibility to different chemoattractants could contribute to host preference in parasitic plants. This is also to be qualified by the authors, especially for obligatory parasites, for which sensitivity to germination stimulants is a major parameter of host preference.

P15L9-10. "but the asymmetrical PIN2 accumulation occurs only in nitrogen-, especially ammonium ion-, deficient conditions." This sentence is a bit too fast shortcut of the results, and needs to be corrected. Indeed, the referenced experiment (Fig.3) does not specifically assess the effect of nitrogen but the effect of a nutrient-rich environment. The effect of ammonium is well shown specifically on the chemotropism and the expression of SMAX1.2, but not on the spatial distribution of PIN2. Please, rectify.

P15 L13-16. may be "a bit brutal" as the end of the discussion. Moreover, it would then be interesting (expected) to place the chemotropism of ammonium towards host roots (here rice) in the context of the negative control of the host-nitrogen parasite, encompassing the effect on the production of SLs and their perception by the parasite.

P16L1-2, "Such future studies will help design solutions for protecting agricultural fields from nuisance weeds." "Help"? this requires clarification. Please specify how this work will help agronomists and breeders in the development of new solutions? (if not those of new varieties that do not produce or produce less strigolactones (germination stimulants), a solution currently being worked by breeders).

Methods

P18L7. Specify the age of *striga* germinated seeds. P18L17. Why this 30° threshold? P19L18. D10 instead of d14.

P21L12. Why is PjUBC2 used as a reference gene? What does it justify? (data not presented in the manuscript).

P23L16. What generation of transgenic *arabidopsis* is used in this study? Are the phenotype and development of plants affected (photos available in additional results?) A little more information on these new transformants would be appreciated.

Legends of figures

P39L1-2 (fig.4). "Coloured boxes indicate gene expression levels of PjKAI2d in roots or *S. hermonthica* KAI2d in seeds, seedlings, or rice plants at 1-, 3-, 7-day post infection (dpi)". It's confusing: it is not well understood in which organs and in which plants ? please clarify.

P39 (Fig.4). Specify the meaning of #1,#9 ... Transgenic lines?

To Reviewer #1:

The Authors studied the novel chemoattractive feature of strigolactones (SLs) in plants. Although SL-induced chemotropism in fungi is a more or less explored phenomenon, this present paper is the very first genuine report in plants. It is presented in an excellent way that natural SLs, unnatural analogues and agonists trigger chemotropism in two hemiparasitic Orobanchaceae genus. I would like to especially praise the use of rice d10 (MAX4) mutant to demonstrate SL-dependent chemotropism in the presence of host plant. The Authors also found that non-starving *P. japonicum* seedlings, non-parasitic Orobanchaceae, as well as *Arabidopsis* are not responsive to SLs in this context, therefore they concluded that this function of SLs is restricted to nutrient-starving hemiparasites. The Authors also found evidence that chemotropism in *P. japonicum* is mediated by auxin responses and the divergent members of KAI2d clade.

My overall impression is that the manuscript is generally well written and the literature of this field has been cited appropriately and in the right context. The experiments have been performed accurately with an adequate number of replicates and evaluated with the proper statistical method. The „Results” and „Discussion” section, as well as the „Methods” are presented in a clear fashion. Supplementary materials, including the videos are all OK. I can't see any flaws regarding the methodology, experimentation and data interpretation presented in this paper.

Response: We appreciate your kind support for publication of our work and constructive suggestions to improve it. According to your comments, we have revised the manuscript as described below:

I have a few suggestions and some comments on the MS. These are:

-line 91: the unnatural enantiomer (-)-5DS (or ent-5DS) signals through the „canonical” KAI2 in *Arabidopsis* (Scaffidi et al., 2014). *P. japonicum* is unresponsive to this compound, highlighting that SLs and not KAI2-ligands trigger chemotropism. I suggest that the Authors should comment on this in the text.

Response: Thank you for your suggestion. We have added the description on this point in the *Discussion* section (lines 263-264).

-line 116 and onward: Authors cannot directly compare chemotropism data obtained from water-agar and 1/2MS+sucrose plates, because two factors were changed simultaneously. Later it became clear that macronutrients obviously inhibit chemotropism in the presence of sucrose, hence, mineral nutrients and not the carbon source/sugar signaling is responsible for the halting chemotropism. I suggest that the Authors should rewrite these sentences to make it clear that the conclusion were not drawn from the direct comparison.

Response: Thank you for pointing out. We have rephrased the text to clearly state that MS mineral nutrients and sucrose are two different things and to avoid direct comparison between the data obtained on water agar and those obtained on 1/2MS with sucrose (lines 119-121 and 126).

-Figure 2h and k: when ionic forms are referred in the text, please use NH₄⁺ and NO₃⁻.

Response: Thank you for suggestion. We have corrected the figure accordingly.

-line 132: downstream SL signaling has been investigated with monitoring PjSMAX1. In *Arabidopsis*, SMAX1 expression is primarily regulated through KAI2. I understand that SL-specific transcriptional changes are weak in most cases and one cannot expect spectacular fold changes upon SL treatment. Why did the authors use SMAX1 and wonder whether they gave a try with other reported SL-specific genes from dicots such as SMXL7, TCP1 or PAPI (Wang et al., 2019), or something else?

Response: Thank you for your indication. In *Arabidopsis*, SMAX1 functions as a repressor of KAI2-mediated signalling rather than SL signalling. As you pointed out, SMXL7 is reported as an SL-responsive gene. However, KAI2d, Orobanchaceae parasite-specific receptors of exogenous SLs, are derived from KAI2 rather than D14, as shown in Fig. 4a or Conn *et al.* (doi: 10.1126/science.aab1140). Therefore, we considered it proper to use SMAX1, a homologue of the repressor of KAI2 signalling. In addition, Li *et al.* recently showed that SMAX1 is also functional in the downstream of SL signalling (doi: 10.1016/j.xplc.2022.100303). To test the effect of SMAX1 on chemotropism to SLs, we also generated PjSMAX1.2-overexpressing hairy roots. As a result, PjSMAX1.2-overexpressing roots did not exhibit chemotropism to *rac*-strigol (Supplementary Fig. 7; lines 140-141). We hence thought it reasonable to investigate SMAX1 as a downstream gene of SL signalling.

Conn, C.E. *et al.* Convergent evolution of strigolactone perception enabled host detection in parasitic plants. *Science* **349**, 540-543 (2015).

Li *et al.* The strigolactone receptor D14 targets SMAX1 for degradation in response to GR24 treatment and osmotic stress. *Plant Commun.* **3**, 100303 (2022).

-line 151 and onward: again, my concerns are more or less the same as in line 116. I suggest to

-1. include 1/2MS agar and water-agar+sucrose data, or

-2. omit 1/2MS+sucrose data from the figure and move it to a supplementary figure (this half of the figure does not add too much for the context, I think), or

-3. extensively explain in the text that not sugar, but the micronutrient/macronutrient constitution of the plates affects chemotropism.

Response: Thank you for giving us three options. We selected the option 3 and explained in the text that not sucrose but MS mineral(s) affect chemotropism (lines 151-152).

-line 193 and onward: real-time PCR data of the PjKAI2 dominant negative mutant expression level should be included, although I do not stick to it. Also, on Figure 4, a photograph of a non-responding transformant should be included (in a similar fashion as on Figure 1.).

Response: Thank you for your indication. It is good to quantify the expression levels of *PjKAI2d2*. However, we decided not to include the expression levels, because we cannot distinguish the native and transgenic expression of *PjKAI2d2*, the difference between which is only a single nucleotide. Instead, we have included the data using the transgenic plants overexpressing the native *PjKAI2d2* to show that chemotropism to *rac*-strigol was impaired in *PjKAI2d2* dominant-negative mutants (Fig. 4d,e). In addition, we have added photographs of SL-responsive and -irresponsive transgenic plants (Fig. 4d,e; Fig. S10). Methods were also added (lines 205-206, 208-210, 421-429, and 776-777).

-line 183: please change PjKAI2d to PjKAI2d2.

Response: Thank you. We corrected the text accordingly (line 191).

-line 288: sources of the *P. japonicum*, *L. philippensis* and *S. hermonthica* seed batches should be presented.

Response: Thank you for your suggestion. We have added sources of *P. japonicum*, *S. hermonthica* and *L. philippensis* seed batches (lines 306, 314 and 322).

-line 318 and onwards: the solid media with nutrients contained 0.8% (w/v) agar, while the media without nutrients contains 0.7% (w/v) agar. What is the reason for this?

Response: Thank you for pointing this out and sorry for the lack of our explanation. We have once tried the solid media with nutrients containing 0.7% (w/v) agar, but due to the difference in the ionic concentration, the hardness was lower than that of the media without nutrients containing 0.7% (w/v) agar. Therefore, we used the solid media with nutrients containing 0.8% (w/v) agar to match hardness. We have added the explanation to this section (lines 341-343).

-Also in this section, the following sentence has been repeated four times, with only the species name changed: "germinated seedlings were transferred carefully to solid medium without nutrients (0.7% INA agar (w/v), pH 5.8). Filter paper disks (4-mm diameter) were soaked with a chemical-containing

solution, before being placed 3-mm from the seedlings.". Please use only one sentence with listing all the species on which the method has been applied. E.g.: "For assays using *P. japonicum*, *S. hermonthica*, *L. philippensis* and *Arabidopsis*, germinated seedlings were transferred...and so on.

Response: Thank you. We apologise for the redundancy of the text. We have rephrased the text according to your suggestion (lines 337-349).

-line 341: please use "fluorescent stereo microscopy" instead of "fluorescence".

Response: Thank you. We have corrected the text accordingly (line 354).

-line 412: "dummies" means oligonucleotide linkers/modules? Please clarify.

Response: Sorry for the confusion. They are dummy modules. We clarified it in the text (line 431).

To Reviewer #2:

The MS by Ogawa et al. describes the interesting results: 1) Chemotropism of *P. japonicum* to SLs; 2) Inhibition of chemotropism to SLs by ammonium; 3) Correlation between chemotropism and auxin. However, solid evidences in the MS are not enough to clarify the underlying mechanism regulating chemotropism. In reaching the required standard I feel that there is a serious omission in the experimental data provided.

Response: Thank you for carefully reading our manuscript. We have answered all of your comments as described below:

Major comments:

1) Genetic evidences are really required to support the connection between tropism and ammonium. What's about the chemotropism of *PjSMAX1.2* loss- and gain-of-function lines in response to ammonium ions?

Response: Thank you for your suggestion. We generated *PjSMAX1.2*-overexpressing plants to test chemotropism to SLs and found that they do not exhibit chemotropism to *rac*-strigol, even on the ammonium-deficient condition (Supplementary Fig. 7; lines 140-141). Our data suggest that *PjSMAX1.2* negatively regulates chemotropism to SLs and exogenous SLs induce the negative feedback by activating *PjSMAX1.2*-mediated signalling, although we cannot conclude the relationships between chemotropism and ammonium ions from our new data.

2) Genetic evidences are also required to support the connection between tropism and auxin. What's about the tropism of pin loss- and gain-of-function lines in response to ammonium ions? And, what's auxin and PIN distribution in dominant-negative plants PjKAI2d2R183H and PjSMAX1.2 loss-/gain-of-function lines?

Response: Thank you for your suggestion. As you pointed, genetic evidence to support the connection between chemotropism to auxin response would improve this manuscript. However, we consider it extremely difficult to perform genetic analyses at present in *P. japonicum* as we can only do root-transient transformation, but not stable transformation. We once treated *P. japonicum* seedlings with naphthylphthalamic acid (NPA), an auxin transport inhibitor, to test the effect of auxin flux on chemotropism. However, NPA greatly inhibited root growth itself, making it impossible to evaluate the effect on chemotropism. Hence, we consider it beyond our scope in this manuscript to perform genetic analyses to investigate the connection between chemotropism to auxin. Once a stable transformation method is established, these genetic analyses should be performed.

3) Chemotropism is related with ammonium and auxin. In order to draw the pathway of SLs regulated chemotropism, authors should clarify the connection between ammonium and auxin. Again, genetic evidences should be provided.

Response: Thank you. It is an important suggestion. However, as described above, it is extremely difficult to perform genetic analyses to elucidate the connection between ammonium and auxin at present. We consider the analyses beyond our scope in this manuscript. We would like to follow your suggestion once a stable transformation method is established.

4) In *Arabidopsis*, PIN2 and PIN3 are important for gravitropism. What about PIN2 trafficking and PIN3 polarization during chemotropic response of *P. japonicum*?

Response: Thank you for your question. We analysed PIN2 localisation for the following reasons: first, we found that exogenous SLs were perceived by the epidermal cells of the root elongation zone (Fig. 1i); second, only *PIN2* was expressed in the root epidermal cells among PINs in *P. japonicum* (doi: 10.1242/dev.187781). Since expression of *PIN3* was not observed, we assume that *PIN3* in *P. japonicum* is not a major contributor to tropism unlike *PIN3* in *Arabidopsis*. In addition, as *PIN2* is a transporter, we would rather think that investigation of trafficking of auxin response, which can be traced by *DR5* expression (Fig. 3a-h), is important.

Wakatake, T., Ogawa, S., Yoshida, S. & Shirasu, K. An auxin transport network underlies xylem bridge formation between the hemi-parasitic plant *Phtheirospermum japonicum* and host *Arabidopsis*. *Development* **147**, dev187781 (2020).

5) Ammonium uptake could induce acidic stress. Does different pH affect chemotropism? Ammonium regulated chemotropism is dependent on pH?

Response: According to your suggestion, we have tested chemotropism to SLs at a pH of 7. We obtained a similar result with the assays at a pH of 5.8 (Supplementary Fig. 5), indicating that the mechanism by which ammonium ions suppress chemotropism to SLs is independent of pH (lines 129-131 and 339). We have added the new data.

6) Exogenous SLs are perceived as indicated by YLG-based fluorescence. It's good to also directly measure endogenous strigolactone contents in roots of *P. japonicum*.

Response: Thank you for your suggestion. The endogenous SLs in *P. japonicum* are also quite interesting. However, structures of the SLs in *P. japonicum* have not been identified, and it is not even known whether *P. japonicum* biosynthesises SLs. If endogenous SLs in *P. japonicum* are identified in the future, it is worth measuring them. In addition, we focused on host-derived SLs as molecular clues to initiate chemotropism rather than endogenous SLs. We used YLG as it is a nice tool to visualise recognition of exogenous SLs.

7) Regarding to the biological significance, it's better to have field test to compare the effect of *P. japonicum* to the host plants under abundant and limited nutrient conditions.

Response: Thank you for your suggestion. It is true that field tests might give us biological significance in natural settings. Regrettably, however, it is difficult to conduct those tests for the following reasons: first, we do not currently have fields to conduct those experiments; second, evaluation of the effects of nutrient conditions on tropism to host plants in the field is likely to be complex, because *P. japonicum* can complete its life cycle with/without the host; third, different nutrient conditions dramatically affect growth of both *P. japonicum* and host, making it much difficult to compare the effects of nutrient conditions on host tropism. We therefore consider field tests beyond our scope in this manuscript.

Minor comments:

1) Proposed model illustrating SLs acting as navigators for parasite roots should be provided.

Response: Thank you for your suggestion. We have provided a scheme of the SL-mediated tropism in Fig. 5 (lines 273 and 779-780).

2) The statistical test (t-test) is not appropriate for multiple comparisons in some graphs. ANOVA analysis should be applied.

Response: We assume that you mean Figure 4b. We made comparisons between conditions for each gene. Since it is quite difficult to separately show the results of ANOVA analysis of each gene in a single panel, we consider it proper to apply *t*-test. On the other hand, according to your suggestion, we have applied ANOVA analysis for the results shown in Fig. 4d,e. Note that we have added the data using the transgenic plants overexpressing the native *PjKAI2d2*.

To Reviewer #3:

The authors have made a remarkable discovery of chemo-attraction of weed roots towards strigolactones (SLs) exuded by nearby host roots. The SL-response is detected by KAI2d SL receptors, associated with auxin asymmetry and inhibited by ammonium. Unlike some other parasitic weeds, *Phtheirospermum japonicum* is a parasitic weed that does not require SLs to trigger germination near a host root, and does not necessarily need a host. It also contains SL receptor genes. Evident question are; is this a real biological phenomenon, how does a weed know when a host is nearby so that it can attack, what is the mechanism of the tropism and does that mechanism involve SLs and auxin? A broader implication is that by understanding such a mechanism, we might be able to figure out how to exploit that knowledge to disrupt weed attack and save crops.

The authors acquired the following results:

Weed roots showed dramatic growth towards a range of natural SL compounds in soaked filter paper disks.

Weed root growth towards host root was not as good a rice mutant that fails to produce SLs.

Ammonium suppressed weed root growth towards the host, perhaps suggesting that the weed may sense ammonium as a signal that host colonisation is not beneficial for survival in nutrient rich soil.

Increase in auxin reporter and asymmetrical increase in YLG and PjPIN2 reporter proteins suggest that the signal differentially affects one side of the root, and auxin causes the bending towards SLs. (Auxin and PIN2 are well-known root tropism regulators.)

PjKAI2d2 and PjKAI2d3.2 showed relatively high expression in basal conditions and were further

increased by SL treatment, making them candidates for perception in weed root cells of host SLs.

PjKAI2d and PjKAI2d3.2 were found to be functional as SL receptors when transformed into *Arabidopsis* and tested for germination stimulation by exogenous SLs.

It was not feasible to create transgenerational japonicum transgenics of edit or knockdown multicopy genes. Therefore, a dominant negative version of PjKAI2d2, PjKAI2d2-R183H, driven by a constitutively active promoter was made, which would be expected to interfere with and counteract endogenous KAI2d function. It disrupted weed chemotropism, suggesting that KAI2d SL receptors in weeds facilitate chemotropism response. PjKAI2d2-R183H also caused feedback on the endogenous weed SL pathway, suggesting the construct had a repressive function.

The study is convincing. Is it novel? The authors note the recent study that *Orobanche cumana* respond to costunolide for chemo-attraction (Krupp et al., 2021). Costunolide looks chemically similar SLs or karrikins. So it is important for the field to know that *Phtheirospermum japonicum* is responding to (natural) SLs rather than costunolide. In future, it would be good to see if *Phtheirospermum japonicum* responds to other compounds like costunolide (did the authors test that?) and to see how extensive the diversification in weed species is, for the chemo-attraction compound response.

In future, it would also be fascinating to figure out how SL-related compounds and KAI2d transmit the signal to promote auxin asymmetry in weed roots, and thus causes root bending. As the authors mentioned, other studies have shown that SLs (and karrikins) can repress auxin transport and interfere with the auxin-inhibition of PIN protein endocytosis, which does not require de novo protein production, etc. However, the cellular mechanism remains unknown. There seems to be a specific effect of AtKAI2 on gravitropism and PIN2 (Swarbreck et al., *Plant Journal*, 2019; Villaecija-Aguilar et al., *PLOS Genetics*, 2019; Villaécija-Aguilar et al., *Current Biology*, 2022).

Response: We appreciate your deep understanding of our manuscript and supportive comments. According to your comments, we have edited the manuscript as described below:

Suggested edits

L85 “rac-strigol (STR)” Sometime too many abbreviations make things harder to read. And I’ve never seen it written STR before. Would it be simpler if you just write rac-strigol? It is not a long word. (It’s ok to keep it in the figures.)

Response: Thank you for your suggestion. As you pointed out, we should have avoided using unnecessary abbreviations for easy reading. In addition, the abbreviation “STR” for strigol has not

been generally used. Therefore, we now have spelled out “*rac*-STR” and “(+)-STR” as “*rac*-strigol” and “(+)-strigol” throughout the text except for figures, respectively.

L111 “SLs are essential for host tropism”. Could you please clarify this statement in the manuscript? Could there be other signals, but SLs override them?

Response: Thank you for your suggestions. We agree with you. As we do not know if there are other signals, so we have thought that “important” seems more appropriate than “essential”. We have changed the text accordingly (line 108).

L143 & 149 Do you want to state that the asymmetry was towards the attractant, here and elsewhere?

Response: Thank you for your clarification. We would like to state as you pointed. We have changed the text to indicate that asymmetry was towards *rac*-strigol (lines 149-153 and 157-159).

L230 “it is tempting to think”. Could the mechanism have evolved in different ways related to nitrogen-deficient conditions? For instance, there is growing literature on the specific root effects of ammonium. This is a recent review on the topic (Jia and von Wirén, J Exp Bot., 2020). Ammonium has been observed to disrupt root growth and gravitropism tropism in non-weed species, and associated with inhibition of PIN auxin transport. It has been suggested that ammonium can be somehow toxic to roots, but I suspect that ammonium acts as a signal to switch from root foraging, in nutrient poor soils, to root slowing and proliferating in a nutrient rich soil zone. Is it possible that SLs/karrikins/KAI2-ligand and ammonium converge on the same cellular mechanism of auxin transport inhibition, and that the *Phtheirospermum japonicum* lineage has co-opted this endogenous SLs/karrikins/KAI2-ligand and ammonium root response mechanism to enhance host interactions? The supplementary video suggests that the weed root actually slows down and thickens upon touching a potential host root, which would fit with SLs acting to repress auxin transport? Something similar can happen with NPA-treated *Arabidopsis* roots. Perhaps this is a simpler alternative concept, particularly if AMF is not present in *Phtheirospermum japonicum*? However, I agree that a problem with this idea is that SL-, KL-like signals, NPA or ammonium, do not appear to induce chemo-attraction in *Arabidopsis*? Does this mean there is some additional mechanism in the weed to make the epidermis PIN2 hyper-sensitive? Would the authors like try something like NPA or ammonium in a filter disk?

Response: Thank you for your scholarly discussion and introduction of the important review by Jia and von Wirén (2020). It is possible that ammonium-mediated suppression of *PIN2* expression and subsequent tropism might have converged in both parasitic and non-parasitic plants. Referring to your discussion and the review by Jia and von Wirén, we have modified texts in the *Discussion*

section (lines 285-290).

To Reviewer #4:

The ability to recognize a host plant is crucial for obligate parasitic plants and benefit for the facultative forms. This recognition implies allelochemicals including germination stimulants (often strigolactones for obligate forms), chemoattractant compounds directing the parasite's growth towards its host and compounds inducing the formation of the haustorium (HIFs), the organ that binds the parasite to its host. So far, while much work has focused on identifying germination stimulants and HIFs and deciphering induced signalling, the existence of host chemotropism, via strigolactones, was strongly suspected but still to be demonstrated, what Ogawa and his collaborators have achieved in this manuscript. In addition, the results presented here also lead to an interesting discussion by the authors concerning (1) the implication of this chemotropism in the preference of hosts in parasitic plants, (2) the diversion by parasitic plants of this perception system originally active in plants to promote mycorrhizal symbiosis, and thus (3) the potential place of mychorization, in addition to host plants, in the mineral nutrition of parasitic plants. In this context, the results presented by Ogawa and his collaborators represent a major advance in the understanding of plant-plant interactions of a parasitic order and draw very interesting research perspectives.

Thus, the results presented in this manuscript demonstrate (1) the positive chemotropism of exogenous strigolactones (natural and synthetic) and of the host (rice) towards the growth of the root (*Pterospermum*) or the radicle (*Striga*) of the parasitic plant; (2) the asymmetrical perception of strigolactones by host root epidermal cells; (3) the involvement in *Pterospermum* of KAI2d proteins (KAI2d2, KAI2d3.2) in strigol perception, leading to asymmetrical overexpression of PIN2 (auxin efflux transporter) in roots. Some results also suggest the specificity of strigolactone chemotropism in parasitic Orobanchaceae since no strigol chemotropism has been detected in a parasitic Orobanchaceae (*Lindenbergia philippensis*) and *Arabidopsis* (autotrophic plant). The inhibitory effect of nitrogen nutrients (ammonium especially) on the chemotropism of strigol towards *Pterospermum* roots is also demonstrated. This effect expresses in post perception of strigol, with the overexpression of SMAX1, a negative regulator of strigolactone signalling. For this, the authors have developed a chemotropism test of chemical compounds and host plants towards young plants of *Pterospermum* (facultative parasite) and germinated seeds of *Striga hermonthica* (obligate parasite). The use of the fluorescent synthetic strigolactone (YLG) has made it possible to visualize the perception of strigolactones by root epidermic cells and thus to discern the effect of chemotropism inhibitors (nitrogen) on receptors (KAI2d) or induced signalling. The involvement of strigolactones and KAI2d receptors in host chemotropism has been validated by using negative non strigolactone d10 rice mutants and dominant *Pterospermum* hairy roots over-expressing a mutated KAI2d protein. The work and the manuscript are of high quality. Nevertheless, some questions remain. Some points deserve clarification to improve the manuscript before publication. My remarks and questions are

listed below:

Response: Thank you for carefully reviewing our manuscript and providing helpful indications for improvement. According to your suggestions and comments, we have revised the manuscript as described below:

Introduction

P3L8. Reference 7 is of historical importance but seems inappropriate here to illustrate the relatively widespread involvement of strigolactones in the germination of obligate root-parasitic plants.

Response: Thank you for your indication. We have replaced Cook *et al.* (1966) with the recent review by Mutuku *et al.* (2021), in which the contribution of SLs to the germination of obligate parasites was described (line 36).

P3L11 (end of paragraph). I agree, but I think it is useful to quote, here in this introduction part, the previous work on the positive chemotropism of lactone sesquiterpenes towards the obligate parasite *Orobanche cumana*, because of its pioneering character.

Response: Thank you for suggestion. As you pointed out, the previous work about chemotropism to lactone sesquiterpenes in *O. cumana* is an important study. Therefore, we cited Krupp *et al.* (2021) in this paragraph (line 37).

P4L13. Ref 27-29. Need to include the work recently published in Plant Comm on PrKAI2d in the obligate parasite *Phelipanche ramosa* (De Saint Germain *et al.* 2021 in Plant Comm.). Integration of these data into the KAI2 phylogenetic tree proposed by the authors should be also required (updating of the data).

Response: Thank you for the nice suggestion. We have added amino acid sequences of KAI2 in *Phelipanche ramosa* (PrKAI2d1-4 and PrKAI2c) to the KAI2 sequence group and updated the phylogenetic trees (lines 443-445, Fig. 4a, Supplementary Figure 8).

Results

P5L15-16. Please specify the reasons for choosing these three natural strigolactones for the study. This is likely useful for people who are not specialists in parasitic plants.

Response: Thank you for your advice. We selected these three strigolactones (SLs) because they represent each type of the structure of SLs. In addition, these three SLs are commercially available

from Olchemim (<https://www.olchemim.cz/Products.aspx?idc=1&idp=30>), which means that any researcher can use these SLs without skill-demanding processes such as chemical synthesis.

P6L3. 100 nM (+)-5DS induces a significantly different tropism on figures 1g (about 50%) and 1h (only 20%). Thus, for the same treatment, the results differ from one set of experiments to another. How do you explain this? Does this question the reliability and robustness of the chemotropism test and therefore the results?

Response: Thank you for carefully reviewing the results. Sorry for the lack of explanation. Our chemotropism assays are highly affected by the detailed experimental conditions such as the amount of moisture on the surface of the medium, which cannot be aligned perfectly. Hence, we set the humidity at 70% for experiments in *P. japonicum* to minimise deviation of results due to humidity (we have added the description to the *Methods* section: line 314). However, it does not mean that the results are not deviated between different experiments. Thus, it is reasonable that the percentage of the plants showing chemotropism was different between different sets and the reliability and robustness of our chemotropism assay is ensured.

P6L2-3. The chemoattractivity of (+)-5DS was comparable to that of (+)-2'-*epi*-5DS: indeed at 1 μ M, but not at 100 nM. So, is the activity of these two SLs really comparable?

Response: Thank you for your comment. As shown in Fig. 1h, alphabets were assigned to each data according to the result of ANOVA analysis. Different letters indicate statistical significance. The data of (+)-5DS and (+)-2'-*epi*-5DS at 100 nM were assigned “ab” and “bc”, respectively. “b” was assigned to both data, indicating that these data were not statistically significant. Thus, the chemoattractivity of (+)-5DS and (+)-2'-*epi*-5DS was comparable at both 1 μ M and 100 nM.

P6L7 (Fig. 1i): YMG 100 μ M for Fig. 1i. Is it correct? (much more concentrated than for tropism tests (Fig. 1f,g)).

Response: Thank you for the indication. We used 100 μ M YLG solution for confocal microscopy because 1 or 10 μ M YLG solution was not sufficient for the clear fluorescent signals. We confirmed in advance that 100 μ M YLG solution induces chemotropism of *P. japonicum* without any side effects such as growth penalties.

P6L8-9 “indicating that asymmetrical SL recognition activity induces an asymmetrical root elongation pattern, leading to chemotropism.” Here, “indicating” seems too affirmative because the authors do not demonstrate the causal link between asymmetric recognition of chemoattractant and chemotropism. This requires correction.

Response: Thank you for your suggestion. We have changed “indicating” to suggesting” to avoid being too affirmative (line 93).

P6 L11-13. “We found that *S. hermonthica* exhibited chemotropism to rac-STR and had an asymmetrical YLG recognition pattern like *P. japonicum*”. According to Fig. 1m, this statement seems abusive. Indeed, the asymmetrical pattern of YMG recognition is sincerely not evident in the experiment with *Striga* germinated seeds. Please, comment this major result concerning the asymmetric status of strigolactones perception, which is assumed by the authors.

Response: Thank you for pointing this out. We apologise for showing a potentially misleading figure. We considered the fluorescence pattern in Fig. 1m asymmetrical because there did not seem to be much YLG-derived fluorescence on the opposite side of the arrowhead. However, as you pointed out, there was a fluorescence on the right side of the root. To clarify that the YLG recognition pattern in *S. hermonthica* is asymmetrical, we repeated experiments. We have obtained new figures and replaced Fig. 1m with a representative figure that clearly showed the asymmetrical fluorescence. Note that the size of the *S. hermonthica* seedlings was larger than that of the previous one, probably because we used a different lot of the seeds and the duration of the incubation before germination was different (still 1 to 2 weeks in both cases).

P6 L14-18 Additional results, notably with a wider panel of strigolactones are needed to assume the idea that chemotropism of strigolactones would be specific to parasitic plants. Are these data available? (supplementary data?)

Response: Thank you for your suggestion. As you indicated, additional results with a larger variety of SLs and plant species would support our hypothesis that chemotropism to SLs might be limited to Orobanchaceae parasitic plants. However, we agree with you that we cannot conclude this with our dataset so we have changed the sentence in line 66. “Chemotropism to SLs was also observed in *Striga hermonthica*, but not in non-parasitic plants, suggesting that this strategy is conceivably -> potentially Orobanchaceae parasite-specific.”

P7L13-14 (Fig. 2d). In contrast to macronutrients (+ sucrose), micronutrients are tested without sucrose, why? This requires clarification.

Response: Thank you for your indication. Sucrose is not included in the components of MS nutrient minerals, but we counted sucrose as a macronutrient, as well as KH_2PO_4 , KNO_3 , NH_4NO_3 , CaCl_2 and MgSO_4 . We have rephrased the text to clearly state that MS mineral nutrients and sucrose are two

different things, and that not sucrose but MS components affected chemotropism to SLs (lines 119-121 and 126).

P8L6-8 (Fig. 2i-k). Data on the effect of nutrients (Fig. 2j,k) on PjSMAX1 expression are missing. Why?

Response: Thank you for suggestion. We apologise for the lack of explanation. Except for the initial experiments shown in Fig. 2i, we only analysed *PjSMAX1.2* because *PjSMAX1.1* lacks several amino acid sequences conserved in other SMAX1/SMXL2 proteins in Lamiales and hence might not be functional. We have added the description in the text and the alignment of SMAX1/SMXL2 family proteins in dicots (lines 138-140; Supplementary Fig. 6).

P9L14. It would be appreciated if Fig.4a was updated with recent KAI2d data in *P. ramosa* (De Saint Germain et al. 2021 in Plant Comm.).

Response: Thank you for your suggestion. As described above, we have added amino acid sequences of KAI2 in *Phelipanche ramosa* (PrKAI2d1-4 and PrKAI2c) to the KAI2 sequence group and updated the phylogenetic trees (lines 443-445, Fig. 4a, Supplementary Figure 8).

P10L5-10. “from previous transcriptome data”. Please indicate references. Indeed, the discrepancy in results between previous and present analyses is regrettable. The explanation given by the authors is plausible. Is a verification possible, has it been done?

Response: Thank you for your suggestion. We have referenced Kurotani *et al.* (2020) in this sentence (line 179). Such discrepancies can occur depending on the similarity of the gene sequences, as we are sometimes unable to distinguish expression levels of the genes with high similarity in transcriptome analyses. As *KAI2d* genes in *P. japonicum* have high similarity (Supplementary Fig. 9), we performed RT-qPCR with the specific primers, which target the coding sequence and the 3'-untranslated region of each gene, to specifically quantify the expression levels. We verified the expression patterns of the *KAI2* genes in *P. japonicum* and found a discrepancy in the expression pattern of *PjKAI2d3* with the transcriptome data (Fig. 4b). In cases such discrepancies occur, qPCR results prevail over transcriptome data.

P10L17. PrKAI2d2 instead of PrKAI2d

Response: Thank you for your correction. We have changed the text accordingly (line 191).

P11L16 (Fig.4e). SMAX1.2 expression in control plants increases in response to strigol. Is an expected result? Comments are required.

Response: Thank you for your indication. We expected the result from Fig. 2i, which we revealed SL-enhanced expression of *PjSMAX1.2* in the wild-type seedlings. We have changed the sentences accordingly to state this expectation (lines 208-210).

Discussion

P12L6. Is the reference 6 appropriate?

Response: Thank you for your suggestion. We assume that you mean P3L6. As you indicated, citation of Yoshida *et al.* (2016) here is not appropriate. Therefore, we have deleted the reference here (line 34).

P12L8-10. Are KAI2 data available for *L. philippensis*? if so, can they support the author's hypothesis?

Response: Thank you for your question. *L. philippensis* has *KAI2c* and *KAI2i* (included in the phylogenetic tree in Supplementary Fig. 8), but not *KAI2d* genes. Although their transcriptional data are not available, we think the absence of the *KAI2d* genes supports our hypothesis.

P12L10-15. Has the chemotropic effect of costunolide been tested by the authors against *Pterospermum* and *striga*? These tests should be necessary to support that susceptibility to different chemoattractants could contribute to host preference in parasitic plants. This is also to be qualified by the authors, especially for obligatory parasites, for which sensitivity to germination stimulants is a major parameter of host preference.

Response: Thank you for your suggestion. According to your suggestion, we have tested the chemotropic activity of costunolide, karrikin 1 and karrikin 2, exogenous compounds whose chemical structures are similar to SLs, using *P. japonicum* and *S. hermonthica*. As a result, none of the compounds showed chemotropic activity for *P. japonicum* and *S. hermonthica* at 1 μ M concentration (Supplementary Fig. 3), at which SLs showed chemotropic activity for both of the hemiparasites (Fig. 1). These data might represent host preferences in *Orobanchaceae* hemiparasites by using SLs but not similar compounds as chemoattractants. We have added the sentences to describe the results and the discussion (lines 108-111 and 230-231).

P15L9-10. "but the asymmetrical PIN2 accumulation occurs only in nitrogen-, especially ammonium ion-, deficient conditions." This sentence is a bit too fast shortcut of the results, and needs to be

corrected. Indeed, the referenced experiment (Fig.3) does not specifically assess the effect of nitrogen but the effect of a nutrient-rich environment. The effect of ammonium is well shown specifically on the chemotropism and the expression of SMAX1.2, but not on the spatial distribution of PIN2. Please, rectify.

Response: Thank you for pointing this out. We have corrected the sentence not to overstate our results (lines 285-288).

P15 L13-16. may be “a bit brutal” as the end of the discussion. Moreover, it would then be interesting (expected) to place the chemotropism of ammonium towards host roots (here rice) in the context of the negative control of the host-nitrogen parasite, encompassing the effect on the production of SLs and their perception by the parasite.

Response: Thank you for your indication. We have modified the texts to further describe the relationships between ammonium ions and PIN2 in both parasitic and non-parasitic plants (lines 288-290).

P16L1-2, “Such future studies will help design solutions for protecting agricultural fields from nuisance weeds.” “Help”? this requires clarification. Please specify how this work will help agronomists and breeders in the development of new solutions? (if not those of new varieties that do not produce or produce less strigolactones (germination stimulants), a solution currently being worked by breeders).

Response: Thank you for your comment. It would be nice if we could specify the solutions for protecting agricultural field. However, future studies have yet to be designed and we have not had any result. Therefore, it is difficult for us at present to clarify how this work will help agronomists and breeders in the development of new solutions, because it is just a speculation. Instead of specification of the solutions, we have changed the sentence to “Such future studies will help -> may encourage design solutions...” not to haste the conclusion (line 300).

Methods

P18L7. Specify the age of striga germinated seeds.

Response: Thank you for your comment. We have already described the age of *S. hermonthica* in the *Plant materials and growth conditions* chapter (lines 314-318).

P18L17. Why this 30° threshold?

Response: Thank you for your question. Prior to chemotropism assays shown in this manuscript, we performed preliminary experiments to set up the experimental conditions. As we tested 61 *P. japonicum* seedlings chemotropism to 0.1% (v/v) DMSO, the angles were between 30° and -30°. We therefore set the threshold 30°. We are able to include the data in the Supplementary file upon request.

P19L18. D10 instead of d14.

Response: Thank you for your correction. We have changed the text accordingly (line 371).

P21L12. Why is *PjUBC2* used as a reference gene? What does it justify? (data not presented in the manuscript).

Response: Thank you for your question. *PjUBC2* is a gene encoding Polyubiquitin C 2 (UBC2) in *P. japonicum*. Due to its ubiquity, the ubiquitin genes have been widely used as references to normalise expression levels of the genes of interest. *PjUBC2* has already been used as a reference gene (doi: 10.1073/pnas.1619078114; doi: 10.1093/plphys/kiaa001). We therefore used *PjUBC2* as a reference as well in this manuscript. To clarify that we used *PjUBC2*, we have revised the sentence in this chapter (lines 403-404). We have provided source data with this manuscript, including Ct value of the *PjUBC2* gene.

Spallek, T. *et al.* Interspecies hormonal control of host root morphology by parasitic plants. *Proc Natl Acad Sci U S A* **114**, 5283-5288 (2017).

Ogawa, S. *et al.* Subtilase activity in intrusive cells mediates haustorium maturation in parasitic plants. *Plant Physiol* **185**, 1381-1394 (2021).

P23L16. What generation of transgenic arabidopsis is used in this study? Are the phenotype and development of plants affected (photos available in additional results?) A little more information on these new transformants would be appreciated.

Response: Thank you for your comment. We used T2 seeds selected on the medium containing hygromycin. We have added the information about generation to the sentence (line 454). We did not observe effects of the transgenes on the growth phenotype at the time point of the germination assays. We hence did not take photos of the seedlings. Since we consider information about developmental phenotype except germination rate is not necessary to discuss germination phenotype, we do not have a plan to take photos. Thank you for your suggestion again.

Legends of figures

P39L1-2 (fig.4). “Coloured boxes indicate gene expression levels of *PjKAI2d* in roots or S.

hermonthica KAI2d in seeds, seedlings, or rice plants at 1-, 3-, 7-day post infection (dpi)”. It’s confusing: it is not well understood in which organs and in which plants ? please clarify.

Response: Thank you for your suggestion. To clarify the description, we have changed the sentence to “...gene expression levels in roots of *P. japonicum* for *PjKAI2d* or in seeds, seedlings, or rice-infecting plants at 1-, 3-, 7-day(s) post infection...” (lines 761-762).

P39 (Fig.4). Specify the meaning of #1,#9 ... Transgenic lines?

Response: Thank you for your suggestion. We have added the sentence “Two independent lines were selected and tested for each transgene” to the figure legend (line 768).

We also made several minor changes in the manuscript as described below:

Discussion: In addition to Jia and von Wiren (2020), We have added the references describing ammonium-induced inhibition of *PIN2* expression and the subsequent asymmetric auxin accumulation (line 290).

Zou, N., Li, B., Dong, G., Kronzucker, H. J. & Shi, W. Ammonium-induced loss of root gravitropism is related to auxin distribution and TRH1 function, and is uncoupled from the inhibition of root elongation in *Arabidopsis*. *J Exp Bot* **63**, 3777-3788 (2012).

Liu, Y. *et al.* Ammonium inhibits primary root growth by reducing the length of meristem and elongation zone and decreasing elemental expansion rate in the root apex in *Arabidopsis thaliana*. *PLoS One* **8**, e61031 (2013).

Materials and Methods: We mistakenly forgot to indicate that genes encoding KAI2d3.2 and KAI2d4.2 in *P. japonicum* were automatically translated before addition to the KAI2 sequence group. Therefore, we changed the sentences to “...PjKAI2d1-PjKAI2d5 as queries (with e-values under $1e^{-100}$), were automatically translated to amino acid sequences and added to the KAI2 sequence group” (line 443). Sorry for the mistake.

Acknowledgements: We mistakenly forgot to acknowledge Prof. Abdel G. Babiker (Environment and Natural Resources and Desertification Research Institute, Sudan) for sharing *S. hermonthica* seeds, therefore we have added him to acknowledgements. Our apologies.

Figure legends: We have added the numbers of repeats and the statements “Source data provided” if needed.

Figures 2 and 4: We have corrected “*PjUBC*” to “*PjUBC2*” to further clarify the reference gene.

Supplementary Fig. 2: We mistakenly gave the wrong indication of the scale bars in c and d. The correct indication of the scale bars is 2 mm; therefore, we have corrected the legend. Sorry for the mistake.

Supplementary Fig. 8: In the current classification, *Physcomitrium patens* is more accurate than *Physcomitrella patens* (doi: 10.1105/tpc.19.00828). Hence, we have changed accordingly.

Rensing, S. A., Goffinet, B., Meyberg, R., Wu, S. Z., Bezanilla, M. The Moss *Physcomitrium* (*Physcomitrella*) *patens*: a model organism for non-seed plants. *Plant Cell* **32**, 1361-1376 (2020)

REVIEWERS' COMMENTS

Reviewer #1 (Remarks to the Author):

The Authors provided adequate and appropriate responses to the comments and suggestions and greatly improved the quality of the manuscript. I have no more questions/comments.

Reviewer #2 (Remarks to the Author):

The authors have resolved all my questions and the revised manuscript has been improved significantly.

Reviewer #3 (Remarks to the Author):

The authors have adequately addressed my concerns. I just have a few recommended edits.

L91 "YLG, a fluorogenic agonist". Perhaps explain briefly here that YLG binds to strigolactone receptors and is hydrolysed into fluorescent products, hence it is an indicator of receptor activity.

L106 First mention of d10? Therefore, "dwarf10 (d10)".

L109 You should say "a karrikin (KAR1)" and it would be good to introduce costunolide and perhaps mention that your results show that strigolactones are distinctly different to costunolide. (I guess there is a very small chance D10 could somehow promote costunolide exudation in rice and strigolactones may mimic costunolide? But the negative costunolide result rules out that out.)

L142 "negatively regulates the SL signalling pathway". Perhaps clarify here that SL signalling likely triggers the degradation of PjSMAX1.2, but ammonium may counteract this by promoting PjSMAX1.2 expression. A similar antagonistic relationship has been suggested for cytokinin (Kerr et al., Plant Journal, 2021). Hence, SL perception, as indicated by YLG fluorescence, does not necessarily translate into PjSMAX1.2 repression and chemotropism.

L151 A slightly different syntax may be better; "Given the results shown in Fig. 2, it is reasonable to suggest that MS macronutrients compromise the asymmetric auxin response rather than sucrose."

L289 "resulting in root gravitropism"

Reviewer #4 (Remarks to the Author):

For my part, the modifications made by the authors correctly respond to the various observations I had made on the original version of the manuscript, particularly concerning the results illustrating strigolactone-related chemotropism in *Striga hermonthica* seedlings (Fig. 1m) and the updating of the phylogenetic tree of strigolactone receptors. I appreciated notably that the authors started again their experiments to get a more obvious pictures of the asymmetric distribution of SL reception for chemotropism in *S. hermonthica*. The authors have taken great care to explain the corrections that I think are appropriate.

I would just like to point out one point of the discussion which, in my opinion, although very interesting, is still too speculative to be integrated in this ms. This is the hypothesis that the difference in host preference between the genus *striga* ssp and *orobanche* ssp could be based on the nature of chemoattractants. The results to date are too limited in *Orobanche* ssp (only *O. cumana* which is very

specific since the germination stimulant in the rhizosphere of its unique host (sunflower) is not a strigolactone but the costunolide). This hypothesis to be issued deserves to be further tested with more species of the *Orobanche* and *Phelipanche* geni, especially with species that germinate only in response to strigolactones, and there are many. If the authors nevertheless wish to retain this hypothesis, I think they should also mention germination stimulants and Haustorium inducing factors as possible determinants of host preference: molecular package in the rhizosphere including germination stimulants, chemoattractants and Haustorium inducing factors).

To Reviewer #1:

The Authors provided adequate and appropriate responses to the comments and suggestions and greatly improved the quality of the manuscript. I have no more questions/comments.

Response: We appreciate your peer-review and helpful comments for publication.

To Reviewer #2:

The authors have resolved all my questions and the revised manuscript has been improved significantly.

Response: Thank you for reviewing our manuscript again. We appreciate your supportive comment on the publication of our work.

To Reviewer #3:

The authors have adequately addressed my concerns. I just have a few recommended edits.

Response: We appreciate your helpful comments. According to your comments, we have revised the manuscript as described below:

L91 “YLG, a fluorogenic agonist”. Perhaps explain briefly here that YLG binds to strigolactone receptors and is hydrolysed into fluorescent products, hence it is an indicator of receptor activity.

Response: Thank you for your suggestion. We have revised the sentence as follows: “...using YLG, which is an indicator of SL receptor activity by binding to SL receptors and being hydrolysed into fluorescent products...” (lines 92-93).

L106 First mention of d10? Therefore, “dwarf10 (d10)”.

Response: Thank you for pointing it out. We have corrected the sentence accordingly (line 107).

L109 You should say “a karrikin (KAR1)” and it would be good to introduce costunolide and perhaps mention that your results show that strigolactones are distinctly different to costunolide. (I guess there is a very small chance D10 could somehow promote costunolide exudation in rice and strigolactones may mimic costunolide? But the negative costunolide result rules out that out.)

Response: Thank you for your suggestion. As we tested costunolide, KAR1, and KAR2 (Supplementary Fig. S3), we corrected the sentence to: “...as karrikins (KAR1 and KAR2) and costunolide, a sunflower chemoattractant...” (line 110). Regarding the hypotheses that exudation of costunolide is promoted in *d10* and that costunolide mimics SL, we would not mention them in the text, since there has not been any result that supports them.

L142 “negatively regulates the SL signalling pathway”. Perhaps clarify here that SL signalling likely triggers the degradation of PjSMAX1.2, but ammonium may counteract this by promoting PjSMAX1.2 expression. A similar antagonistic relationship has been suggested for cytokinin (Kerr et al., *Plant Journal*, 2021). Hence, SL perception, as indicated by YLG fluorescence, does not necessarily translate into PjSMAX1.2 repression and chemotropism.

Response: Thank you for your scholarly discussion and introduction of cross-talk between SL and cytokinin via a homologue of PjSMAX1.2 (*Plant J* **107**, 1756-1770 (2021)). Referring the possibility that SL signalling triggers the degradation of PjSMAX1.2 protein and that SL signalling via PjSMAX1.2 might be affected by cytokinin signalling, we have added the sentences in the text (lines 145-150) and added the following references:

Li, Q. *et al.* The strigolactone receptor D14 targets SMAX1 for degradation in response to GR24 treatment and osmotic stress. *Plant Commun* **3**, 100303 (2022). (Reference 41)

Kerr, S. C. *et al.* Integration of the SMXL/D53 strigolactone signalling repressors in the model of shoot branching regulation in *Pisum sativum*. *Plant J* **107**, 1756-1770 (2021). (Reference 42)

L151 A slightly different syntax may be better; “Given the results shown in Fig. 2, it is reasonable to suggest that MS macronutrients compromise the asymmetric auxin response rather than sucrose.”

Response: Thank you for your suggestion. We have revised the sentence accordingly (line 159-160).

L289 “resulting in root gravitropism”

Response: Thank you. We have corrected the sentence accordingly (line 299).

To Reviewer #4:

For my part, the modifications made by the authors correctly respond to the various observations I had made on the original version of the manuscript, particularly concerning the results illustrating strigolactone-related chemotropism in *Striga hermonthica* seedlings (Fig. 1m) and the updating of the phylogenetic tree of strigolactone receptors. I appreciated notably that the authors started again their

experiments to get a more obvious pictures of the asymmetric distribution of SL reception for chemotropism in *S. hermonthica*. The authors have taken great care to explain the corrections that I think are appropriate.

Response: Thank you for your careful peer-review of our manuscript. We appreciate your feedback on our revisions.

I would just like to point out one point of the discussion which, in my opinion, although very interesting, is still too speculative to be integrated in this ms. This is the hypothesis that the difference in host preference between the genus *striga* ssp and *orobanche* ssp could be based on the nature of chemoattractants. The results to date are too limited in *Orobanche* ssp (only *O. cumana* which is very specific since the germination stimulant in the rhizosphere of its unique host (sunflower) is not a strigolactone but the costunolide). This hypothesis to be issued deserves to be further tested with more species of the *Orobanche* and *Phelipanche* geni, especially with species that germinate only in response to strigolactones, and there are many. If the authors nevertheless wish to retain this hypothesis, I think they should also mention germination stimulants and Haustorium inducing factors as possible determinants of host preference: molecular package in the rhizosphere including germination stimulants, chemoattractants and Haustorium inducing factors).

Response: Thank you for the important suggestion. To avoid overstating the possibility that differences in host preferences between species might come from the nature of chemoattractants, we have changed the sentences from “...the nature of chemoattractants is different...” to “...the nature of chemoattractants might be different...” (lines 235-236). We have also mentioned that as well as chemoattractants, haustorium-inducing factors might be determinants of host preferences (lines 239-242). Regarding germination stimulants of *O. cumana*, while the effect of host-derived SLs on germination remains unclear, GR24 was shown to function as a germination stimulant (*Science* **349**, 540-543 (2015); *Front Plant Sci* **12**, 699068 (2021)). We therefore hypothesise that chemoattractants or haustorium-inducing factors might be important for host preferences rather than germination stimulants.